*Report*

# The mutational landscape of the SCAN-B real-world primary breast cancer transcriptome

Christian Brueffer[1,2] (ID), Sergii Gladchuk[1,2], Christof Winter[1,2,†] (ID), Johan Vallon-Christersson[1,2,3], Cecilia Hegardt[1,2,3], Jari Häkkinen[1,2] (ID), Anthony M George[1,2], Yilun Chen[1,2], Anna Ehinger[1,2,4], Christer Larsson[2,5], Niklas Loman[1,2,6], Martin Malmberg[6], Lisa Rydén[1,2,7], Åke Borg[1,2,3] & Lao H Saal[1,2,3,*] (ID)

## Abstract

Breast cancer is a disease of genomic alterations, of which the panorama of somatic mutations and how these relate to subtypes and therapy response is incompletely understood. Within SCAN-B (ClinicalTrials.gov: NCT02306096), a prospective study elucidating the transcriptomic profiles for thousands of breast cancers, we developed a RNA-seq pipeline for detection of SNVs/indels and profiled a real-world cohort of 3,217 breast tumors. We describe the mutational landscape of primary breast cancer viewed through the transcriptome of a large population-based cohort and relate it to patient survival. We demonstrate that RNA-seq can be used to call mutations in genes such as *PIK3CA*, *TP53*, and *ERBB2*, as well as the status of molecular pathways and mutational burden, and identify potentially druggable mutations in 86.8% of tumors. To make this rich dataset available for the research community, we developed an open source web application, the SCAN-B MutationExplorer (http://oncogenomics.bmc.lu.se/MutationExplorer). These results add another dimension to the use of RNA-seq as a clinical tool, where both gene expression- and mutation-based biomarkers can be interrogated in real-time within 1 week of tumor sampling.

**Keywords** breast cancer; mutation; RNA-seq; survival; transcriptome
**Subject Categories** Cancer; Chromatin, Transcription & Genomics

## Introduction

Mutations in the cancer genome, including single nucleotide variants (SNVs) and small insertions and deletions (indels), can shed light on cancer biology, tumor evolution and susceptibility or resistance to therapeutic agents (The Cancer Genome Atlas, 2012; Bose *et al*, 2013; Robinson *et al*, 2013). Mutations can now even be used to track circulating tumor DNA in the blood of patients (Garcia-Murillas *et al*, 2015; Förnvik *et al*, 2019). In recent years, the characterization of the mutational landscape of breast cancer has been performed primarily on the DNA level (The Cancer Genome Atlas, 2012; Cheng *et al*, 2015; Ciriello *et al*, 2015). Adoption of massively parallel RNA sequencing (RNA-seq) as a clinical tool has been slower, despite several complementary advantages over DNA-seq. In addition to gene and isoform expression profiling and detection of *de novo* transcripts such as fusion genes, RNA-seq can approximate classical DNA-seq capabilities in the detection of SNVs, indels, as well as structural variants (Ma *et al*, 2018) and coarse copy number (preprint: Talevich & Shain, 2018). This makes RNA-seq an excellent tool for biomarker development (Brueffer *et al*, 2018) and potential clinical deployment (Byron *et al*, 2016; Cieślik & Chinnaiyan, 2018).

For these reasons, among others, in 2010, the Sweden Cancerome Analysis Network–Breast (SCAN-B) initiative (ClinicalTrials.gov ID NCT02306096) selected RNA-seq as the primary analytic tool (Saal *et al*, 2015; Rydén *et al*, 2018). SCAN-B is a prospective real-world and population-based multicenter study with the aim of developing, validating, and clinically implementing novel biomarkers. To this end, SCAN-B collects tumor tissue and blood samples from enrolled patients with a diagnosis of primary breast cancer (BC). To date, over 15,000 patients have been enrolled, and messenger RNA (mRNA) sequencing is performed on patient tumors within 1 week of surgery. All patients are treated uniformly according to the Swedish national standard of care regimen.

Expression profiling is an excellent tool to develop gene signatures for established and novel biomarkers (Sotiriou *et al*, 2006; Roepman *et al*, 2009; Brueffer *et al*, 2018), and many such signatures can be applied to a single RNA-seq dataset. However, for the detection of SNVs and indels from RNA-seq data, there are several challenges.

1  Division of Oncology, Department of Clinical Sciences, Lund University, Lund, Sweden
2  Lund University Cancer Center, Lund, Sweden
3  CREATE Health Strategic Center for Translational Cancer Research, Lund University, Lund, Sweden
4  Department of Pathology, Skåne University Hospital, Lund, Sweden
5  Division of Molecular Pathology, Department of Laboratory Medicine, Lund University, Lund, Sweden
6  Department of Oncology, Skåne University Hospital, Lund, Sweden
7  Department of Surgery, Skåne University Hospital, Lund, Sweden
   *Corresponding author. Tel: +46 46 2220365; E-mail: lao.saal@med.lu.se
   †Present address: Institut für Klinische Chemie und Pathobiochemie, Klinikum rechts der Isar, Technische Universität München, München, Germany

Unlike DNA-seq, where whole-genome or targeted sequencing reads are distributed approximately uniformly and in proportion to DNA copy number, the abundance of reads in RNA-seq is proportional to the expression of each gene or locus. Consequently, only variants in expressed transcripts of sufficient level can be detected. In cancer, this means that variants in oncogenes can likely be detected, whereas those in tumor suppressor genes, e.g., *TP53*, *BRCA1*, or *BRCA2*, are more likely to be missed. For example, mutations inducing premature stop codons can lead to nonsense-mediated decay, causing loss of expression and subsequently false-negative calls. The transcriptome is also more complex and challenging than the genome. RNA structures, such as alternative splicing, add computational challenges to alignment, and RNA editing can contribute to false-positive variant calls. Finally, there is the lack of benchmark datasets for RNA-seq, as are available for DNA from the Genome in a Bottle consortium and others (Zook *et al*, 2016; Li *et al*, 2018).

The aim of this study was to optimize RNA-seq somatic mutation calling through comparison to matched targeted DNA-seq, discern the mutational landscape of the early breast cancer transcriptome across a large cohort of 3,217 treatment-naïve SCAN-B cases with sufficient follow-up time, and to make the resulting vast dataset available for exploration by the wider research community. To demonstrate the power of the methodology and dataset, we assessed the impact of mutations in important breast cancer driver genes and pathways, as well as tumor mutational burden (TMB) on patient overall survival (OS).

# Results

An outline of the study design, which comprised DNA sequencing and RNA sequencing of 275 samples from the ABiM cohort, and RNA sequencing of 3,217 samples from the SCAN-B cohort, is shown in Fig 1.

## Variant filter performance

Mutation calling in the 275 sample ABiM cohort resulted in 3,478 somatic post-filter mutations from the matched tumor/normal targeted capture DNA, and 1,459 variants from tumor RNA-seq in the DNA capture regions (Table 1 and Fig EV1A). Comparing these DNA and RNA variants resulted in 1,132 mutations that were present both in DNA and RNA in the capture regions and whose frequencies were generally in line with previous studies such as The Cancer Genome Atlas (TCGA) (The Cancer Genome Atlas, 2012) (Fig EV1B). Of the 1,459 RNA-seq variants, 884 (60.6%) were identified as somatic in DNA, 248 (17.0%) as germline in DNA, and 327 (22.4%) as unique to RNA. These RNA-unique variants are a mix of somatic mutations missed in DNA-seq, e.g., due to regional higher sequencing coverage in RNA-seq or tumor heterogeneity, unfiltered RNA editing sites, or artifacts caused by PCR, sequencing, or alignment and variant calling.

## Landscape of somatic mutations in the breast cancer transcriptome

We applied the filters derived from the 275 sample set to the entire RNA-seq SCAN-B 3,217 sample set, resulting in 144,593 total

variants comprised of 141,095 SNVs, 1,112 insertions, and 2,386 deletions (Table 1). The number of mutations per sample in the SCAN-B set was lower compared to the ABiM set, likely due to the ABiM set being sequenced to a higher depth (Table EV1). The SNVs comprised 50,270 missense, 2,311 nonsense, 1,042 splicing, 68,819 affecting 3′/5′ untranslated regions (UTRs), 17,057 synonymous mutations, as well as 1,596 mutations predicted otherwise. The majority of indels were predicted to cause frameshifts or affect 3′/5′ UTRs (Table EV2). After removing synonymous mutations, the number of mutations was reduced to 127,536 variants in the SCAN-B set, i.e., an average of 40 mutations per tumor.

We analyzed the contribution of the six nucleotide substitution types (C>A, C>G, C>T, T>A, T>C, and T>G) to SNVs in the ABiM and SCAN-B sets (Fig 2A). Compared to DNA, RNA-seq-based variant calls showed a relative under-representation of C>T substitutions and an over-representation of T>C substitutions.

In accordance with published studies of primary BC, the most frequently mutated genes were the known BC drivers *PIK3CA* (34% of samples), *TP53* (23%), *MAP3K1* (7%), *CDH1* (7%), *GATA3* (7%), and *AKT1* (5%) (Fig 3). As reported before (Ciriello *et al*, 2015), disruptive alterations in *CDH1* were a hallmark of lobular carcinomas (135/386 [35.0%] of samples), while alterations in *TP53*, *MAP3K1*, and *GATA3* were more common in the ductal type. 86.8% of SCAN-B samples had at least one mutation in a gene targeted by an approved or experimental drug, based on the Database of Gene-drug Interactions (DGI).

## Somatic mutations in important BC genes

We examined known driver BC genes more closely and found our RNA-seq-based mutation calls to recapitulate known mutation rates and hot spots, summarized in Table 2, Table EV2, and Fig 2C–F. Associations of mutated genes and clinical and molecular biomarkers are summarized in Table EV3, and several examples are highlighted below.

*PIK3CA* was the most frequently mutated gene, with 1,163 nonsynonymous mutations in 1,095 patient samples (34% of patients). As expected, and in line with previous studies (Saal *et al*, 2005; The Cancer Genome Atlas, 2012; Pereira *et al*, 2016), the majority of alterations were the known hot spot mutations H1047R/L, E545K, and E542K (Table 2, Fig 2D), which lead to constitutive signaling (Bader *et al*, 2006). All hot spot mutations and the vast majority of other *PIK3CA* alterations were missense mutations. Mutations were associated with lobular, ER$^+$, PgR$^+$, HER2$^-$, and Luminal A (LumA) BC (Table EV3).

*TP53* is frequently disrupted by somatic SNVs; however, a few hot spot mutations exist (Giacomelli *et al*, 2018). The mutation frequency in BC is estimated to be 35.4-37% (The Cancer Genome Atlas, 2012; Pereira *et al*, 2016), which we could confirm in our DNA-seq ABiM filter-definition cohort (37%). Likely due to nonsense-mediated decay (NMD), loss of heterozygosity, and/or decreased mRNA transcription, in the 3,217 cases, the frequency of *TP53* mutations was lower at 23% (782 mutations in 733 samples). Despite underdetection by RNA-seq, the identified hot spot residues were the same as reported in the IARC TP53 database (release R20) (Bouaoun *et al*, 2016). The most often mutated amino acids we observed were R273, R248, R175 (50, 49, and 24 mutations respectively, total 123/782 [15.7%]), followed by positions Y220 (21/782

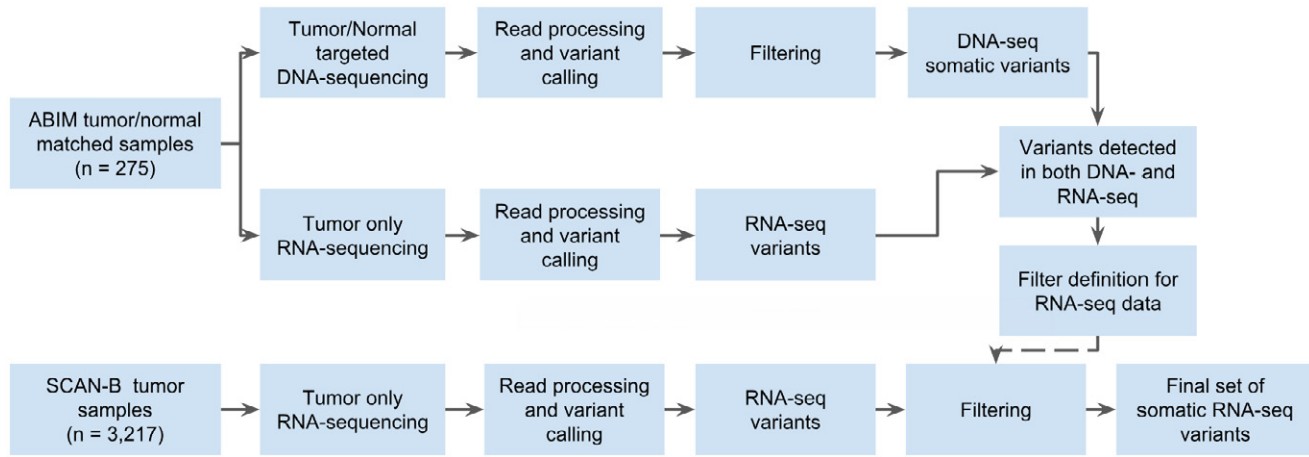

**Figure 1. Study design.**
Study design flow diagram for DNA-seq-informed optimization of RNA-seq variant calling.

**Table 1. Number of mutations in the ABiM (DNA-seq and RNA-seq) and SCAN-B (RNA-seq) cohorts.**

| Cohort | Source | Coverage | Total mutations | SNVs | Insertions | Deletions | Samples with mutations | Mutations per sample |
|---|---|---|---|---|---|---|---|---|
| ABiM | DNA | Capture regions | 3,478 | 3,173 | 50 | 173 | 274 | 12.7 |
| ABiM | RNA | Capture regions | 1,459 | 1,304 | 57 | 98 | 265 | 5.5 |
| ABiM | RNA | Whole mRNA | 16,683 | 15,764 | 235 | 684 | 275 | 60.7 |
| SCAN-B | RNA | Whole mRNA | 144,593 | 141,095 | 1,112 | 2,386 | 3,217 | 44.9 |

Sample numbers differ from total cohort sizes due to filtering resulting in samples with no remaining post-filter mutations.

[2.7%]), R280 (19/782 [2.4%]), and R342 (17/782 [2.2%]) (Table 2, Fig 2C). Most detected mutations are in the DNA binding domain, and 77.6% of overall mutations are missense mutations, likely leading to protein loss of function (LoF). As anticipated, *TP53* mutations were associated with ductal, ER$^-$, PgR$^-$, HER2$^+$, hormone receptor positive (HoR$^+$)/HER2$^+$ (HoR$^+$ defined as ER$^+$ and PgR$^+$, HoR$^-$ otherwise), HoR$^-$/HER2$^+$, triple-negative BC (TNBC), and the basal-like and HER2-enriched PAM50 subtypes (Table EV3), as reported before (The Cancer Genome Atlas, 2012).

*PTEN* is a crucial tumor suppressor gene and regulator of PI3K activity, and PTEN protein expression is associated with poor outcome (Saal *et al*, 2007). In our dataset, we found 124 non-synonymous mutations in 116/3,217 (3.6%) samples, including hot spot mutations in H303 and H266 of unknown significance (Fig 2E). Mutations were significantly associated with HER2$^-$ disease (Table EV3).

*ERBB2* (HER2) mutations have emerged as a novel biomarker and occur by the majority in patients without *ERBB2* amplification (Bose *et al*, 2013), but also in *ERBB2*-amplified cases (Cocco *et al*, 2018). Evidence is mounting that recurrent *ERBB2* mutations lead to increased activation of the HER2 receptor in tumors classified as HER2 normal (Bose *et al*, 2013; Wen *et al*, 2015; Pahuja *et al*, 2018). Activating *ERBB2* mutations have been shown to confer therapy resistance against standard of care drugs such as trastuzumab and lapatinib (Cocco *et al*, 2018), but can be overcome using pan-HER tyrosine kinase inhibitors (TKIs) such as neratinib (Bose *et al*, 2013; Ben-Baruch *et al*, 2015; Ma *et al*, 2017; Cocco *et al*, 2018).

*ERBB2* mutations have also been shown to confer resistance to endocrine therapy in the metastatic setting (Nayar *et al*, 2018), where HER2-directed drugs are effective (Murray *et al*, 2018). We identified 117 non-synonymous *ERBB2* mutations in 103 patients (3.2%), higher than the previously reported incidence rates of 1.6%-2.4% (Bose *et al*, 2013; Wen *et al*, 2015; Ross *et al*, 2016), but lower than in metastatic BC where rates as high as ~7% have been reported (Cocco *et al*, 2018). Two hot spots, L755S (28/117) and V777L (24/117) that cause constitutive HER2 signaling (Fig 2F) (Bose *et al*, 2013; Wen *et al*, 2015), accounted for 44.4% of total *ERBB2* mutations. Co-occurrence of *ERBB2* mutation and amplification has been reported before, however mainly in the metastatic setting (Cocco *et al*, 2018). In our untreated, early BC cohort, we observed *ERBB2* mutation and amplification in 12 tumors, demonstrating that co-incident *ERBB2* mutation and amplification is rare but can occur in early, treatment-naïve BC. Mutation and amplification were not mutually exclusive (*P* = 0.88), and interestingly *ERBB2* mutations occurred predominantly in tumors classified as PAM50 HER2-enriched subtype (*P* = 0.0001). Moreover, *ERBB2* mutation was significantly associated with PgR$^-$ and lobular BC (Table EV3).

Loss of E-cadherin (CDH1) protein expression is a hallmark of the lobular BC phenotype (Ciriello *et al*, 2015). With 12% of our cohort being of lobular type, we observed 137 of total 233 *CDH1* mutations in lobular BCs (58.8%, *P* = 1.6E-72). The mutations were mostly comprised of nonsense mutations (37.2%) and frameshift indels (35.4%), suggesting they contribute to CDH1

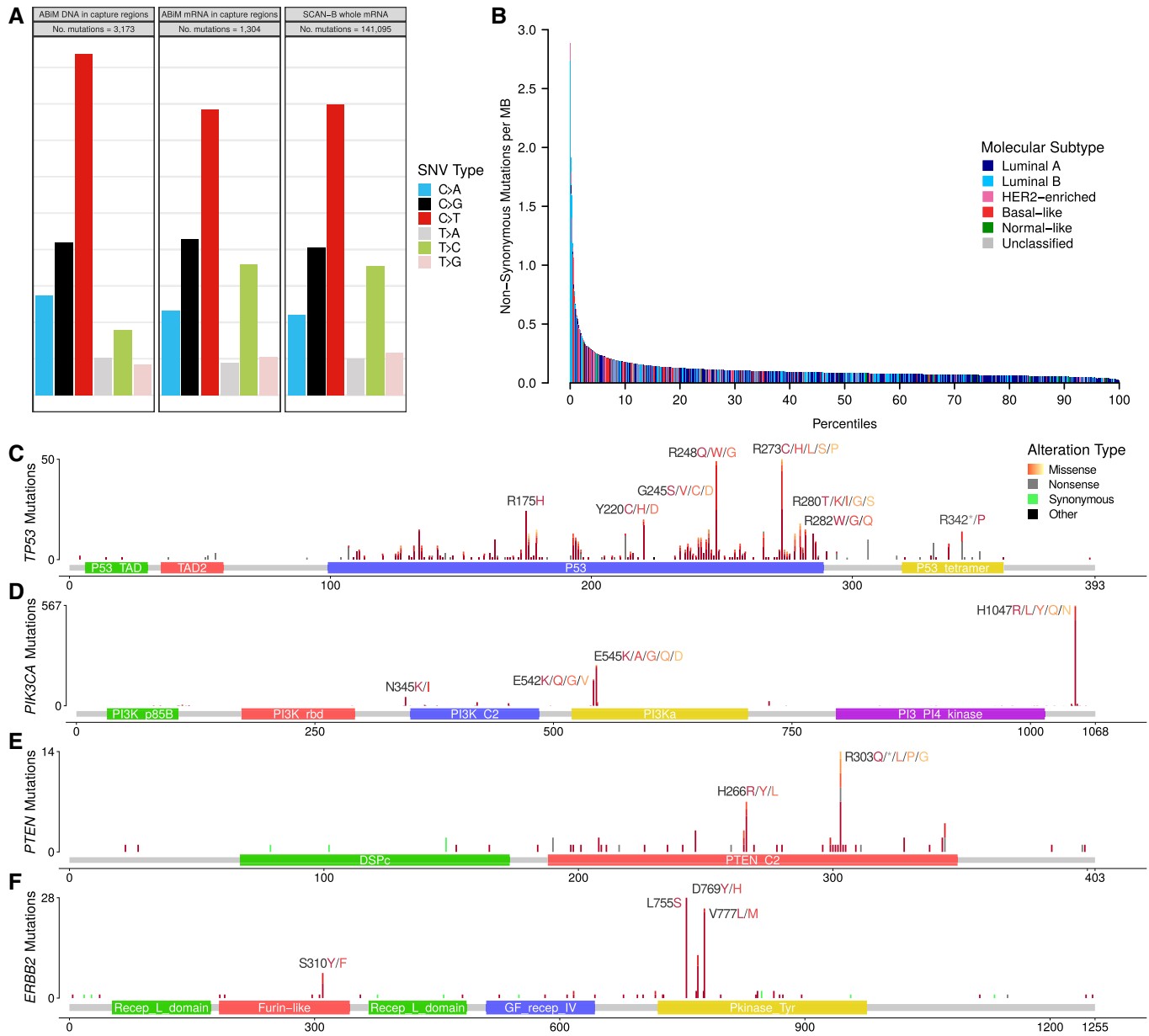

**Figure 2. Overview of non-synonymous mutations in terms of base substitution signatures, molecular subtype, and protein impact.**

A   Contribution of base change types to the overall SNV composition in the ABiM cohort for captured DNA regions and mRNA in the captured DNA regions, as well as SCAN-B whole mRNA.

B   Number of non-synonymous mutations per sample. Bars are colored by PAM50 subtypes Luminal A (dark blue), Luminal B (light blue), HER2-enriched (pink), basal-like (red), Normal-like (green) and Unclassified (gray).

C–F  Lollipop plots showing the location, abundance, and impact of SNVs in (C) *TP53*, (D) *PIK3CA*, (E) *PTEN*, and (F) *ERBB2* on the respective encoded protein. Protein change labels are shown for the most mutated amino acid positions, with residues ordered left to right by mutation frequency within each label.

expression loss and drive the lobular phenotype. We observed one nonsense mutation hot spot (Q23*, *n* = 18), and this residue was also hit by a rare missense mutation (Q23K, *n* = 1). In addition to lobular BC, *CDH1* mutations were associated with ER⁺, HER2⁻, and HoR⁺/ HER2⁻ status, and the LumA subtype (Table EV3).

Other notable mutated genes in our set were *MAP3K1*, *AKT1*, *ESR1*, *GATA3*, *FOXA1*, *SF3B1*, and *CBFB*. *MAP3K1* is a regulator of signaling pathways and regularly implicated in various cancer types.

Loss of *MAP3K1* expression activates the PI3K/AKT/mTOR pathway and desensitizes the tumor to PI3K inhibition (Avivar-Valderas *et al*, 2018), thus mutation status of this gene may affect efficacy of PI3K-targeting drugs. We observed a high rate of frameshift indels, and missense mutations mostly clustered in the kinase domain. Co-mutation of *MAP3K1* and *PIK3CA* occurred in 108 tumors (3.4%), and inactivating (frameshift/nonsense) *MAP3K1* alterations occurred in 77 of 1,095 (7%) of *PIK3CA*-mutant tumors. *AKT1* is a

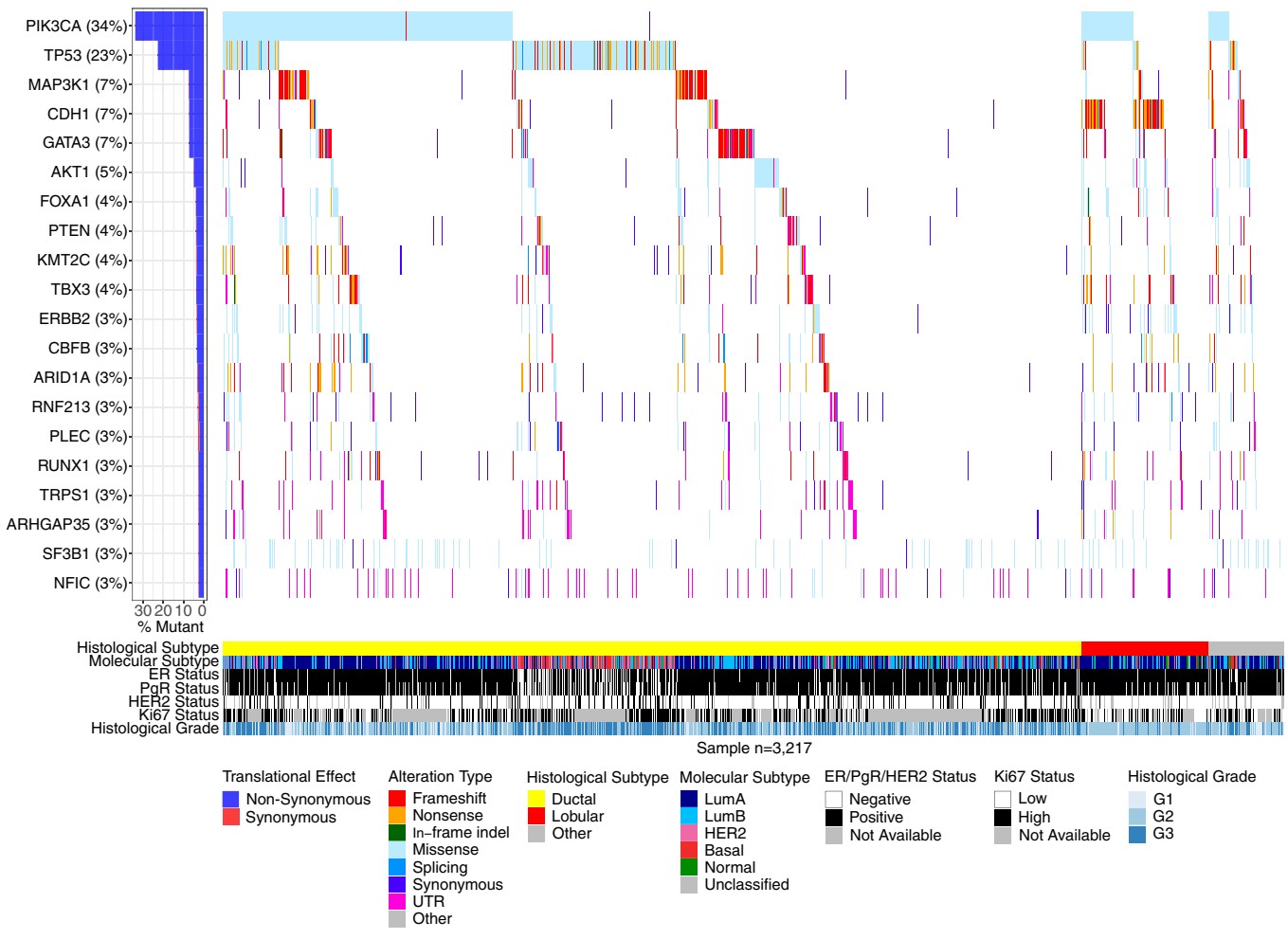

**Figure 3. Overview of frequently mutated genes across 3,217 SCAN-B samples.**

Waterfall plot of the 20 most frequently mutated genes (rows) across 3,217 SCAN-B samples (columns). Genes are ranked from top to bottom by mutation frequency. Samples are sorted by histological subtype and alteration occurrence. Mutations are colored by predicted functional impact.

common oncogene with 156 (4.8%) mutated samples and featured the fourth most mutated hot spot (E17K, 121 mutations) in the SCAN-B cohort. These mutations are predictive of sensitivity to AKT inhibitors (Hyman *et al*, 2017). *ESR1* encodes the estrogen receptor (ER) alpha, perhaps the most important clinical BC biomarker. Seventy-seven tumors harbored 81 *ESR1* variants, including known endocrine treatment resistance mutations, that are discussed elsewhere in detail (M. Dahlgren, AM. George, C. Brueffer, S. Gladchuk, Y. Chen, J. Vallon-Christersson, C. Hegardt, J. Häkkinen, L. Rydén, M. Malmberg, C. Larsson, SK. Gruvberger-Saal, A. Ehinger, N. Loman, Å. Borg, LH. Saal, submitted). Relatedly, *GATA3* and *FOXA1* are frequently mutated transcription factors that are directly involved in modulating ER signaling, and their expression is independently associated with beneficial survival in ER$^+$ tumors (Hisamatsu *et al*, 2012). We identified 246 *GATA3* mutations, including known recurrent frameshift mutations (P409fs, $n = 30$ and D336fs, $n = 10$) and the M294K/R missense mutation ($n = 15$), as well as 10 splice site variants. In *FOXA1*, we detected 146 total mutations, including known recurrent S250F ($n = 23$) and F266L/C ($n = 12$)

missense mutations. Most mutations occurred in the forkhead DNA binding domain. While the role of mutations in these genes has not been thoroughly characterized, Takaku *et al* (2018) suggest that *GATA3* can function as either oncogene or tumor suppressor depending on the mutations the gene accumulated, and which part of the protein product is impacted. According to their classification, the most frequent mutation in our cohort, the P409fs frameshift mutation, results in an elongated protein product compared to *GATA3*-wt that has favorable survival compared to mutations of the second Zinc finger domain. In line with their involvement in ER signaling, mutations in *GATA3*, *FOXA1*, *MAP3K1*, and *ESR1* were associated with ER$^+$ and PgR$^+$ disease. Further, *GATA3, MAP3K1,* and *ESR1* were associated with HoR$^+$/ HER2$^-$, and *GATA3* and *MAP3K1* with ductal BC, while *ESR1* and *FOXA1* were more common in lobular BC. All these genes were associated with the LumA subtype, with the exception of *GATA3* which was associated to Luminal B (LumB) (Table EV3).

*SF3B1* encodes a subunit of the spliceosome and mutations in this gene have been identified as potentially interesting treatment

**Table 2.** The most occurring non-synonymous mutations in the genes *PIK3CA*, *AKT1*, *SF3B1*, *GATA3*, *ERBB2*, *TP53*, *FOXA1*, and *CDH1* in 3,217 SCAN-B samples.

| Gene | AA change | Number of mutations | Mut. samples (%) | Mut. in gene (%) |
|------|-----------|---------------------|------------------|------------------|
| *PIK3CA* | H1047R | 483 | 15 | 41.5 |
| | E545K | 212 | 6.6 | 18.2 |
| | E542K | 142 | 4.4 | 12.2 |
| | H1047L | 77 | 2.4 | 6.6 |
| | N345K | 49 | 1.5 | 4.2 |
| | E726K | 26 | 0.8 | 2.2 |
| | C420R | 20 | 0.6 | 1.7 |
| | E453K | 13 | 0.4 | 1.1 |
| | G1049R | 11 | 0.3 | 0.9 |
| | E545A | 10 | 0.3 | 0.9 |
| | Q546K | 10 | 0.3 | 0.9 |
| | M1043I | 8 | 0.2 | 0.7 |
| | Other | 102 | 3.2 | 8.8 |
| *AKT1* | E17K | 121 | 3.8 | 76.1 |
| | Other | 38 | 1.2 | 23.9 |
| *SF3B1* | K700E | 60 | 1.9 | 74.1 |
| | Other | 21 | 0.7 | 25.9 |
| *GATA3* | P409fs | 30 | 0.9 | 12.2 |
| | M294K | 14 | 0.4 | 5.7 |
| | D336fs | 10 | 0.3 | 4.1 |
| | D332fs | 10 | 0.3 | 4.1 |
| | Other | 182 | 5.7 | 74 |
| *ERBB2* | L755S | 28 | 0.9 | 23.9 |
| | V777L | 24 | 0.7 | 20.5 |
| | D769Y | 9 | 0.3 | 7.7 |
| | Other | 56 | 1.7 | 47.9 |
| *TP53* | R273C | 25 | 0.8 | 3.2 |
| | R248Q | 25 | 0.8 | 3.2 |
| | R175H | 24 | 0.7 | 3.5 |
| | R248W | 22 | 0.7 | 3.1 |
| | R273H | 19 | 0.6 | 2.4 |
| | Y220C | 17 | 0.5 | 2.2 |
| | F134L | 14 | 0.4 | 1.8 |
| | E285K | 13 | 0.4 | 1.7 |
| | R213* | 12 | 0.4 | 1.5 |
| | R282W | 12 | 0.4 | 1.5 |
| | R306* | 10 | 0.3 | 1.3 |
| | Y163C | 10 | 0.3 | 1.3 |
| | L194R | 9 | 0.3 | 1.2 |
| | R342* | 9 | 0.3 | 1.2 |
| | E286K | 8 | 0.2 | 1 |
| | G245S | 8 | 0.2 | 1 |
| | H179R | 8 | 0.2 | 1 |

**Table 2** (continued)

| Gene | AA change | Number of mutations | Mut. samples (%) | Mut. in gene (%) |
|------|-----------|---------------------|------------------|------------------|
| | Q331* | 8 | 0.2 | 1 |
| | Other | 529 | 16.4 | 65.1 |
| *FOXA1* | S250F | 23 | 0.7 | 15.8 |
| | F266L | 11 | 0.3 | 7.5 |
| | Other | 112 | 3.5 | 76.7 |
| *CDH1* | Q23* | 18 | 0.6 | 7.7 |
| | I650fs | 8 | 0.2 | 3.4 |
| | P127fs | 8 | 0.2 | 3.4 |
| | Other | 199 | 6.2 | 85.4 |

Shown are the total number of mutations, the frequency of the mutations in the SCAN-B cohort (Mut. samples), and the frequency of a particular mutation within all mutations in the gene (Mut. in gene).

targets after having been observed in myelodysplastic syndromes and chronic lymphocytic leukemia. We identified 81 *SF3B1* mutations in 79 tumors, 60 of which were K700E hot spot mutations that deregulate splicing and result in differential splicing patterns in BC (Maguire *et al*, 2015). Alterations in this gene are associated with ER$^+$ disease (Maguire *et al*, 2015) and affect alternative splicing patterns (Alsafadi *et al*, 2016). The cohort frequency of 1.9% K700E mutations matches up with previously reported 1.8% in an unselected breast cancer cohort (Maguire *et al*, 2015). We could not confirm the reported prevalence of *SF3B1* mutations in ER$^+$ tumors in the total ER$^+$ group ($P = 0.052$), but in the ER$^+$/ HER2$^-$ subgroup (68/79 mutated tumors ER$^+$/ HER2$^-$, $P = 0.021$), as well as the association with non-ductal, and non-lobular subtypes ($P = 0.0033$). Additionally, *SF3B1* mutations were associated with LumB tumors ($P = 0.0006$) (Table EV3).

CBFB is a transcriptional co-activator of RUNX2, an expression regulator of several genes involved in metastatic processes such as cell migration. Increased CBFB expression has been identified as essential for cell invasion in BC (Mendoza-Villanueva *et al*, 2010). Recurrent *CBFB* mutations have recently been reported in ER$^+$/ HER2$^-$ disease; however, the significance of these mutations is unknown (Griffith *et al*, 2018). We could confirm this finding showing 107 mutations (3.3% cohort frequency), 95 of which were in ER$^+$/ HER2$^-$ samples (4% of ER$^+$/ HER2$^-$ samples, $P = 0.0005$). We also found them to be associated with the LumA subtype (Table EV3); however, we did not observe the splice site mutation described by Griffith *et al* (2018), perhaps due to degradation of the spliced mRNA by NMD.

### Mutations in molecular pathways

We were interested whether the mutational data, when considered from the perspective of mutated pathways, could reveal new biological correlates. To test this, we mapped mutation status to important BC pathways as defined in the Reactome database (Fabregat *et al*, 2018; Jassal *et al*, 2020). We called a pathway mutated when at least one of the member genes had a non-synonymous mutation and clustered samples by pathway mutation status using Euclidean distance and Ward linkage. Notable

clusters that emerged were co-mutated hedgehog signaling, p53-independent DNA repair, and hypoxia response pathways, as well as a cluster of NOTCH1/2/3 signaling mutated tumors, both in mostly basal-like and HER2-enriched tumors. Both clusters are linked in their relation to cancer stem cell development (Habib & O'Shaughnessy, 2016; Locatelli & Curigliano, 2017), which, in addition to the NOTCH and Hedgehog pathways themselves, has emerged as a novel treatment target, particularly in TNBC. Another co-mutation cluster was made up of PI3K/AKT, MET, RET, EGFR, ERBB2, and ERBB4 signaling pathways that occurred in a subset of Luminal A and B tumors (Fig 4; see Table EV4 for Reactome pathway IDs). Activation of these pathways is involved in the development of $ER^+$ BC through proliferation-inducing signaling, or endocrine therapy resistance, e.g., via activating *ERBB2* mutations (Nayar *et al*, 2018).

## Tumor mutational burden

Tumor mutational burden is increasingly of interest due to its association to neoantigen burden and response to immunotherapies. We used the median number of non-synonymous mutations per transcriptome megabase (rnaMB), 0.082 mutations/rnaMB, to stratify all SCAN-B samples into TMB-high and TMB-low groups. Samples with HER2-enriched and basal-like PAM50 subtypes were enriched in the top 10% of samples with the highest TMB compared to the lowest 90% ($P$ = 2.2E-16, Fig 2B), supporting previous results and indicating that immunotherapy may have higher activity in these two PAM50 subtypes (The Cancer Genome Atlas, 2012).

## Mutational landscape and patient outcomes

Next, we were interested in the association between mutations in important BC genes and patient outcome under various treatments. Below we show the results for *TP53*, *PIK3CA*, *ERBB2*, and *PTEN* with OS of SCAN-B patients in clinical biomarker and treatment groups (Figs 5 and EV2), as well as selected pathways (Figs 6 and EV3). Specific treatments stratified by clinical biomarker and treatment groups are detailed in Table EV4. The web tool SCAN-B MutationExplorer may be used to query any gene(s) and pathway(s) of interest.

In line with expectations, *TP53* mutation predicted poor survival in untreated patients (hazard ratio [HR] 2.39, 95% CI [1.5–3.79], $P$ = 0.00014), patients treated with endocrine- and chemotherapy (HR: 1.83 [1.09–3.05], $P$ = 0.02), as well as the $HoR^+/HER2^-$ biomarker subgroup (HR: 1.43 [1.06–1.94], $P$ = 0.019). After adjusting for important covariates in multivariable (MV) Cox analyses, *TP53* mutations remained a significant stratifier among patients receiving endocrine- and chemotherapy.

In early-stage breast cancer, *PIK3CA* mutations have been associated with slightly better 5-year OS than *PIK3CA*-wt tumors in univariable analysis, but not when correcting for clinicopathological and treatment variables (Zardavas *et al*, 2018). In our hands, we saw a similar univariable effect in patients who did not receive systemic treatment (HR: 0.54 [0.32–0.91], $P$ = 0.018), but not when adjusting for covariates. Additionally, *PIK3CA* mutations in HER2 ± any treated patients became significant in multivariable analysis.

*ERBB2* mutations were indicators of poor prognosis in endocrine therapy only (HR: 1.85 [1.08–3.18], $P$ = 0.023) and endocrine- and chemotherapy-treated (HR: 3.49 [1.4–8.72], $P$ = 0.0042) patients, as well as in the $HoR^+/HER2^-$ subgroup (HR: 1.96 [1.14–3.35], $P$ = 0.013). After multivariable adjustment, they remained a significant predictor in the endocrine-only-treated patient subgroup.

*PTEN* mutations alone were associated with poor survival in the patient group not receiving systemic treatment (HR: 2.56 [1.03–6.33], $P$ = 0.036), but not in any of the other treatment or clinical biomarker groups (Fig 5 and EV2). While loss of PTEN protein expression or non-functional PTEN protein can be caused by SNVs and indels, it can also be caused by other mechanisms such as large structural variants (Saal *et al*, 2008) and promoter methylation (Zhang *et al*, 2013) that have not been investigated in this study. To account for this, we defined a new subgroup *PTEN*-MutExp, where a status of "low" identifies cases with either *PTEN* mutation or gene expression in the lower quartile within the cohort, and "normal" otherwise. The *PTEN*-MutExp low group, incorporating gene expression, showed improved stratification in the no systemic treatment group (HR: 1.88 [1.2–2.95], $P$ = 0.0053), and significantly lower OS in patients receiving only endocrine treatment (HR: 1.63 [1.26–2.12], $P$ = 0.00021), as well as $HoR^+/HER2^-$ patients (HR: 1.54 [1.2–1.99], $P$ = 0.00076). Most of the prognostic value is provided by the gene expression, however mutation data improved stratification (Fig EV4). After multivariable adjustment, *PTEN* mutations in the no systemic-treated subgroup, as well as the *PTEN*-MutExp "low" group in $HoR^+/HER2^-$ and $HoR^+/HER2^+$ patients, remained significant.

Abstracting from mutations in individual genes, we investigated the effect of mutated pathways on OS in patient subgroups stratified by treatment (Fig 6) and clinical subgroup (Fig EV3). Mutated WNT (Fig 6A, HR: 2.14 [1.18–3.89], $P$ = 0.01), Hedgehog (Fig 6B, HR: 1.68 [1.06–2.68], $P$ = 0.026), and NOTCH2 (Fig 6C, HR: 2.31 [1.27–4.2], $P$ = 0.0047) pathways, as well as the p53-independent DNA damage repair pathway (Fig 6D, HR: 2.03 [1.3–3.17], $P$ = 0.0015) were associated with worse survival in patients not receiving systemic treatment. Additionally, NOTCH2 signaling (Fig 6C, HR: 1.65 [1.19–2.3], $P$ = 0.0026) was associated with worse OS in patients receiving only endocrine treatment, and $TGF\beta$ signaling (Fig 6E, HR: 1.79 [1.08–2.96], $P$ = 0.021) with worse OS in patients treated with endocrine- and chemotherapy. Further, WNT signaling was associated with worse OS in $HoR^+/HER2^+$ (HR: 2.57 [1.04–6.33], $P$ = 0.034) and TNBC patients (HR: 2.5 [1.27–4.91], $P$ = 0.0061; Fig EV3). In multivariable analysis, WNT pathway mutations in $HoR^+/HER2^+$ and TNBC patients, NOTCH2 pathway mutations in endocrine-only-treated patients, and $TGF\beta$ pathway mutations in endocrine + chemo ± any treated patients remained significant stratifiers.

Given its importance as an emerging biomarker for response to immune checkpoint therapy (Goodman *et al*, 2017), we investigated whether TMB could also provide response information with respect to conventional treatment regimens (Fig 7). When stratified into TMB-high and TMB-low by the SCAN-B cohort median TMB per rnaMB, low TMB was favorable to OS independent of treatment across the cohort (HR, 1.54 [1.28–1.86], $P$ = 0.0000033), as well as in patients not systemically treated (HR: 2.53, [1.58–4.05], $P$ = 0.000066), treated with endocrine therapy only (HR: 1.55 [1.22–

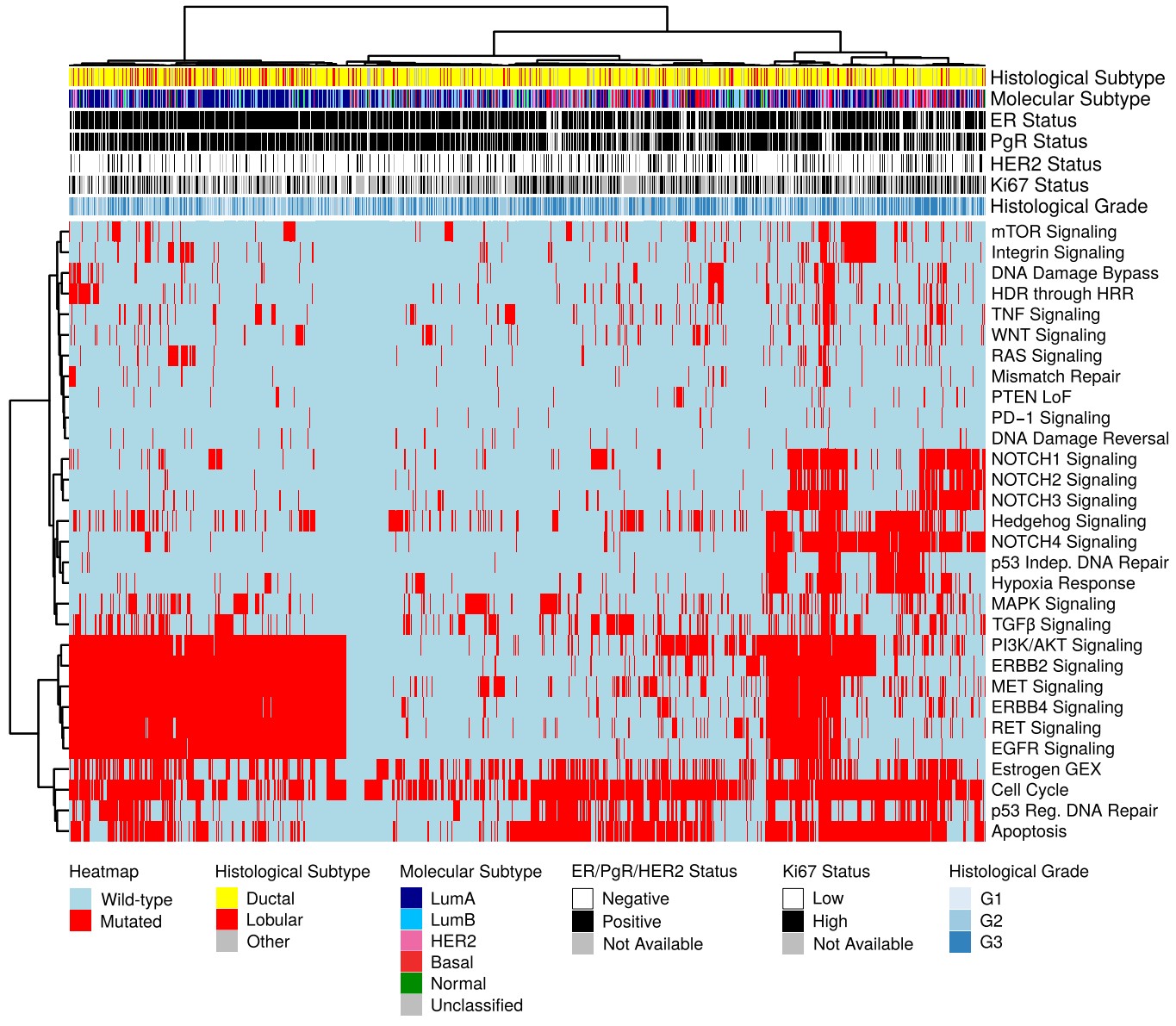

**Figure 4.  Binary heatmap of mutation status of important breast cancer pathways in 3,217 samples.**

Binary heatmap of mutation status of important BC pathways in 3,217 samples. Samples with wild-type (wt) pathway status (defined as all member genes being wt) are colored blue, those with mutated pathways (at least one member gene mutated) are colored red. Samples and pathways were clustered using Euclidean distance and Ward linkage. Reactome IDs for the pathways can be found in Table EV4.

1.98], $P = 0.00036$), endocrine ± any therapy (HR: 1.4 [1.12–1.74], $P = 0.0028$), and chemotherapy ± any therapy (HR: 1.66 [1.12–2.47], $P = 0.011$). High TMB is typically associated with improved survival in TNBC, possibly due to increased neoantigen load enabling a stronger immune response. However, we observed no such effect in TNBC patients within the SCAN-B cohort ($P = 0.34$, Fig EV5). Mutational load was a significant survival stratifier across the Nottingham Histological Grade (NHG) grading scheme (G1, HR: 0.38 [0.16–0.9], $P = 0.022$; G2, HR: 1.46 [1.1–1.94], $P = 0.0078$; G3, HR: 1.53 [1.13–2.07], $P = 0.0055$), and within the ER$^+$ (HR: 1.41 [1.15–1.74], $P = 0.00097$), PgR$^+$ (HR: 1.28 [1.02–1.59], $P = 0.031$), HER2$^-$ (HR: 1.53 [1.25–1.86], $P = 0.000024$), and Ki67-high (HR:

1.76 [1.17–2.65], $P = 0.0064$) patient subgroups (Fig EV5). Interestingly, LumB patients with high TMB showed worse survival (HR: 1.58 [1.13–2.21], $P = 0.0064$), whereas TMB was not a significant stratifier for any other molecular subtype (Fig EV5). LumB tumors were also the only subgroup where TMB remained a significant stratifier in multivariable analysis.

**SCAN-B MutationExplorer**

To enable public exploration and re-use of our rich mutational dataset, we developed the web-based application SCAN-B MutationExplorer (available at http://oncogenomics.bmc.lu.se/MutationExplorer; Fig 8).

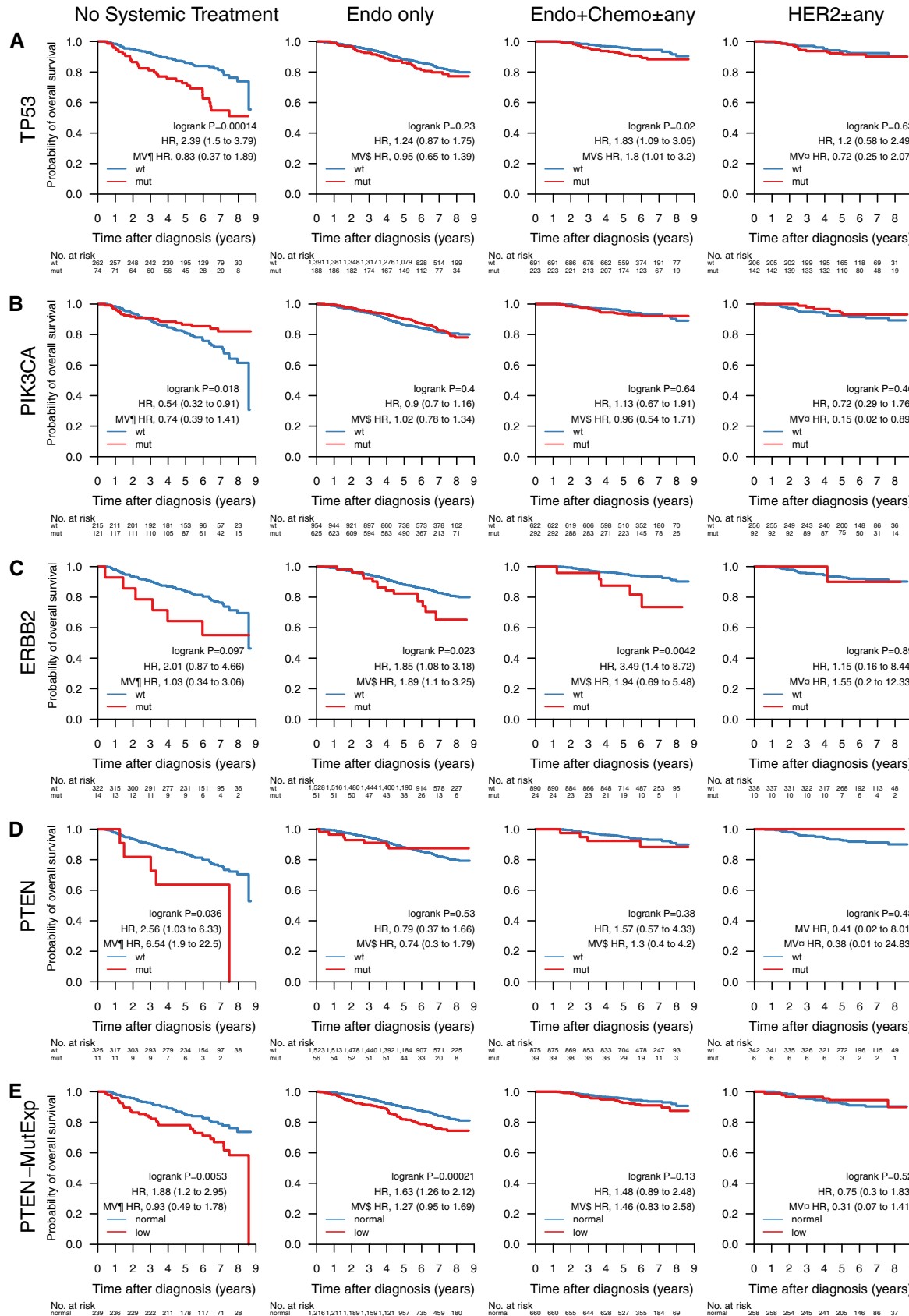

**Figure 5.**

**Figure 5. Impact of gene mutations on overall survival across treatment groups.**

A–E Overall survival (OS) of patients with tumors containing mutations in the genes (A) *TP53*, (B) *PIK3CA*, (C) *ERBB2*, and (D) *PTEN*. (E) OS by *PTEN*-MutExp genotype ("low" defined as *PTEN* mutation or *PTEN* expression in the lower quartile across the cohort, "normal" otherwise) stratified by groups receiving no systemic treatment (n = 336), endocrine therapy only (Endo only; n = 1,579), endocrine- and chemotherapy (Endo + Chemo ± any; n = 914), as well as HER2 treatment with any other treatment or none (HER2 ± any; n = 348). Specific treatments in these groups are detailed in Table EV5. In each Kaplan–Meier plot, wild-type (wt) and normal cases are plotted in blue, mutated (mut) and low cases are plotted in red, the log-rank P value is given, and the hazard ratio (HR) for mutation/low is given with a 95% CI and after univariable and multivariable (MV) Cox regression adjustment. Covariables included in the MV analysis were age at diagnosis, lymph node status, tumor size, and the variables denoted by the following symbols: ¶, ER, PgR, HER2, and NHG; ¤, ER, PgR, and NHG; $, HER2 and NHG. ER, estrogen receptor; HER2, human epidermal growth factor receptor 2; NHG, Nottingham histological grade; PgR, progesterone receptor.

With this interactive application, a user can filter the 3,217 SCAN-B samples based on combinations of clinicopathological and molecular markers (histological type, ER, PgR, HER2, Ki67, NHG, and PAM50 subtype), treatments (endocrine, chemotherapy, HER2 treatment), and mutations based on mutation type (e.g., nonsense or missense) and COSMIC occurrence. From the filtered data, the user can create mutational landscape waterfall plots and conduct survival analysis using KM analysis and log-rank tests based on mutations in single genes, pathways as defined in the Reactome database or custom, as well as TMB, either using the absolute number of mutations, or mutations per expressed MB of genome, using a user-defined threshold. Mutations can also be plotted from a protein point of view using user-defined occurrence cutoffs for showing and annotating mutations. Plots in PDF format as well as the mutation set underlying the currently active plot in tab-separated values (TSV) format can be downloaded for further analysis. The application is based on R Shiny and the source code is available under the BSD 2-clause open source license at http://github.com/cbrueffer/MutationExplorer.

## Discussion

Tumor somatic mutation status is a crucial piece of information for the future of precision medicine to guide treatment selection and give insight into tumor evolution. Analysis of DNA is the gold standard for detecting SNVs, indels, and larger structural variants. However, many interesting tumor properties are only accessible on the transcriptome level and cannot be interrogated using DNA; most prominently gene expression at the isoform and gene level, as well as *de novo* transcripts originating from gene fusions. The SCAN-B initiative (Saal *et al*, 2015) decided early on to perform RNA-seq on the tumors of all enrolled patients. Based on this, we have developed, refined, and benchmarked gene expression signatures (Brueffer *et al*, 2018; Dihge *et al*, 2019; Lundgren *et al*, 2019; Søkilde *et al*, 2019; Vallon-Christersson *et al*, 2019), and detected recurring fusions affecting miRNAs (Persson *et al*, 2017). Herein, we described the development of a pipeline for detection of somatic SNVs and indels based on RNA-seq, adding another layer to information that can now be obtained from a single sequencing analysis within 1 week of surgery (Saal *et al*, 2015).

To date, several approaches for RNA-seq mutation calling, mostly in combination with matched DNA, have been developed (Horvath *et al*, 2013; Piskol *et al*, 2013; Radenbaugh *et al*, 2014; Wilkerson *et al*, 2014; Guo *et al*, 2017; Siegel *et al*, 2018); however, calling from RNA-seq alone, particularly from tumor-only samples, is still a challenge. With the advance of targeted and whole exome sequencing into the clinics, and efforts such as TCGA, MSK-Impact,

and others, variant calling from DNA-seq has improved in recent years, although discordance between detection pipelines still exists (Hofmann *et al*, 2017; Ellrott *et al*, 2018; Shi *et al*, 2018). Part of this improvement is the availability of validation resources such as the Genome in a Bottle datasets (Zook *et al*, 2016). With clinical interest in RNA-seq only recently picking up, e.g., as shown by two recent review articles (Byron *et al*, 2016; Cieślik & Chinnaiyan, 2018), comparably well-characterized RNA-seq datasets for validation do not yet exist to our knowledge.

The strategy for mutation calling herein was to perform initial variant calling with low requirements on coverage and base quality to increase sensitivity while allowing false positives. To increase specificity, we then applied stringent *post hoc* filtering that can be easily amended as further annotation data become available, or as existing sources receive updates. The advantage of this two-step strategy is the possibility to accommodate different research and clinical questions in the future that may have different filtering needs.

Two major contributors of false-positive mutation calls are germline SNPs/indels and RNA editing. Common approaches for dealing with germline events are calling mutations from matched tumor/normal samples, or filtering SNPs present in databases such as dbSNP. The latter is problematic, since some dbSNP entries with a low variant allele frequency (VAF) may be legitimate somatic mutations. On the other hand, filtering on the dbSNP "common" flag (at least 1% VAF in any of the 1,000 genomes populations) can lead to many low-VAF germline SNPs remaining. We tried to address this issue by combining the dbSNP and COSMIC databases, and only filtering variants present in dbSNP if they were not present in COSMIC. We filtered out known RNA editing sites using publicly available databases; however, there is still an overabundance of T>C substitutions in our RNA-based calls compared to DNA-based calls, suggesting many unknown editing sites and insufficient filtering (Fig 2B). Approaches have been developed to identify RNA editing sites using DNA/RNA-trained machine learning models (Sun *et al*, 2016) or RNA-seq data alone (Ramaswami *et al*, 2013), which may provide ways to improve filtering in the future by creating a SCAN-B RNA editing database.

The overall landscape of somatic mutations in our study looked similar to that reported previously from DNA (The Cancer Genome Atlas, 2012; Pereira *et al*, 2016), with the two most frequently mutated genes *PIK3CA* (34% of samples) and *TP53* (23%), followed by other known drivers *MAP3K1* (7%), *CDH1* (7%), *GATA3* (7%), and *AKT1* (5%) (Fig 2). While mutation frequencies in oncogenes such as *PIK3CA* are generally in line with previous reports, frequencies in tumor suppressor genes were generally lower in RNA-seq than would be expected from our study population. For example,

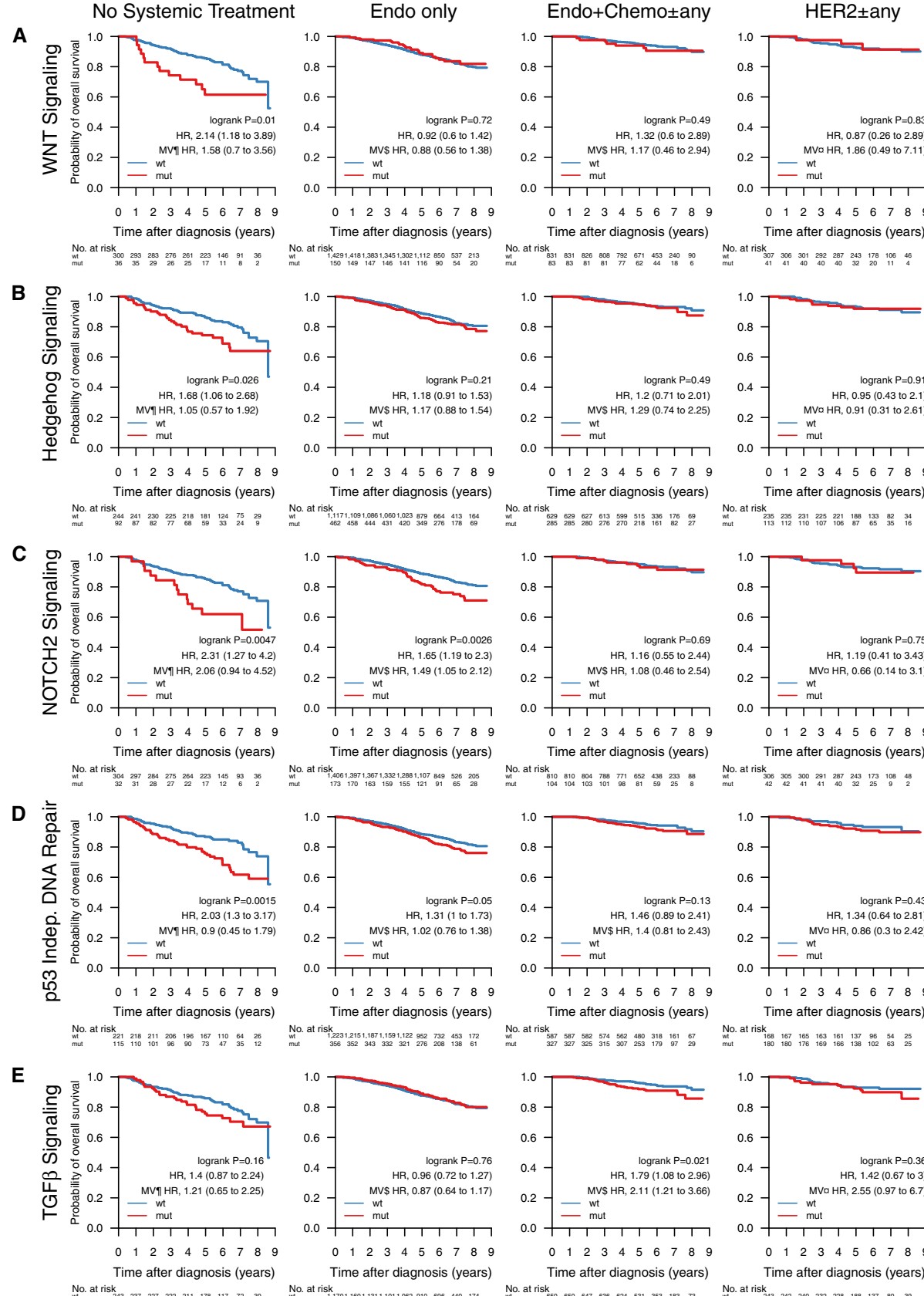

**Figure 6.**

**Figure 6. Impact of pathway mutations on overall survival across treatment groups.**

A–E Overall survival of patients with tumors containing mutations in pathways (A) WNT signaling, (B) Hedgehog signaling, (C) NOTCH2 signaling, (D) p53 independent DNA damage repair, (E) TGFβ signaling, stratified by groups receiving no systemic treatment (n = 336), endocrine therapy only (Endo only; n = 1,579), endocrine- and chemotherapy (Endo + Chemo ± any; n = 914), as well as HER2 treatment with any other treatment or none (HER2 ± any; n = 348). Specific treatments in these groups are detailed in Table EV4. In each Kaplan–Meier plot, wild-type (wt) cases are plotted in blue, mutated (mut) cases are plotted in red, the log-rank P value is given, and the hazard ratio (HR) for mutation is given with a 95% CI and after univariable and multivariable (MV) Cox regression adjustment. Covariables included in the MV analysis were age at diagnosis, lymph node status, tumor size, and the variables denoted by the following symbols: ¶, ER, PgR, HER2, and NHG; ¤, ER, PgR, and NHG; $, HER2 and NHG. See Table EV3 for Reactome pathway IDs. ER, estrogen receptor; HER2, human epidermal growth factor receptor 2; NHG, Nottingham histological grade; PgR, progesterone receptor.

our *TP53* RNA-seq somatic mutation frequency of 23% (reference: 36%, cBioPortal.org) suggests we may be missing a significant fraction of *TP53* mutations present in DNA. Similar trends can be seen in *PTEN* (observed: 3.6%, reference: 4.6%), *BRCA1* (observed: 0.2%, reference: 1.6%), and *BRCA2* (observed: 0.03%, reference: 2.2%). This is not surprising since only mutations in sufficiently highly expressed genomic regions can be detected by RNA-seq and loss of expression of tumor suppressor genes is a hallmark of oncogenesis. Furthermore, truncated mRNAs caused by nonsense mutations are typically removed by nonsense-mediated decay before they can be captured for sequencing. Thus, our findings do not reflect the true mutational spectrum of tumor suppressor genes. Despite these limitations, we could identify a putative mutation in at least one gene targeted by an existing drug in the majority of patient tumors (86.8%), demonstrating that it should be feasible to match most patients to targeted treatments using RNA-seq analyses.

One of the major oncogenic pathways in breast cancer is PI3K/AKT/mTOR, which is frequently upregulated by activating mutations in *PIK3CA*, *MAP3K1*, and *AKT1*, or inactivating mutations in *PTEN*, leading to increased growth signaling. This pathway is being targeted by multiple drugs, such as alpelisib (Novartis) (Juric *et al*, 2018) in HoR⁺/HER2⁻ *PIK3CA* mutant tumors in combination with fulvestrant (André *et al*, 2019), and the AKT1 inhibitor AZD5363 (AstraZeneca) (Hyman *et al*, 2017). The strength of RNA-seq in mutation profiling lies within oncogenes, and we demonstrate that alterations in drug targets such as *PIK3CA* and *AKT*, as well as genes potentially modulating drug efficacy, such as *MAP3K1*, can be detected. Eventually, RNA-seq may be used as companion diagnostic for oncogene-targeting drugs such as these. While we also detected mutations in *PTEN*, these only showed significant prognostic power when combined with low gene expression in the *PTEN*-MutExp low group, suggesting either SNVs and indels are a minor mechanism of PTEN loss in early BC compared to structural rearrangements (Saal *et al*, 2008), and other means of *PTEN* expression loss. Taken together, we detected mutations in multiple PI3K/AKT/mTOR signaling nodes that lead to increased pathway activation and have emerging clinical utility in luminal BC, e.g., through combination with EGFR inhibition as demonstrated in basal-like BC (She *et al*, 2016).

Loss of p53 activity, either through LoF mutations, dominant-negative mutations, or low expression, is a major contributor to tumorigenesis. While RNA-seq generally underdetects *TP53* mutations, the identified hot spot residues remain the same as reported in the IARC TP53 database. Clinically these mutations could already be actionable, as *TP53* mutations are a sign of DNA damage repair deficiency and may be prognostic for sensitivity to PARP inhibition (Holstege *et al*, 2010; Severson *et al*, 2015). Patients with *TP53*-mutant tumors had significantly worse OS in the patient subgroups treated only with endocrine therapy, or no systemic treatment at all (Fig 5), and HoR⁺/HER2⁻ patients (Fig EV2), suggesting that *TP53* mutations identify a subgroup of patients that are spared chemotherapy or systemic therapy overall by appearing low risk, but are in fact high-risk patients that should be treated accordingly.

Endocrine treatment is the most important first-line treatment in BC. Resistance to these treatments leads to disease progression and recurrence and has been studied extensively. Drivers for endocrine resistance include activating mutations in *ESR1* and *ERBB2* which have been studied mostly in the metastatic setting. We show that mutations in these genes already occur in early, untreated BC, with 177 (5.5%) of patients in our population-based cohort having a mutation in either gene. We further demonstrate that patients with these mutations that received only endocrine treatment have inferior OS, suggesting drug resistance. Detecting these patients early could open up additional treatment options that have shown efficacy in the metastatic setting, such as selective estrogen receptor degraders (SERDs) in *ESR1*-mutated tumors, or TKIs such as neratinib in *ERBB2*-mutated BC.

The role of alternative splicing in tumorigenesis has recently garnered increased attention, and the extend of isoform switching in several cancer types, including BC, has been characterized (Vitting-Seerup & Sandelin, 2017). Mutations such as the *SF3B1* K700E hot spot mutation deregulate splicing and result in differential splicing patterns in BC (Maguire *et al*, 2015). The clinical effect of these mutations is unclear, and we did not detect significant survival stratification in important biomarker or treatment groups. However, the fact that mutations in splicing-related genes can be detected from RNA-seq make this method attractive for research and possible clinical use, as they can be correlated with expression originating from the same sequencing experiment.

Individual mutations, particularly in infrequently mutated genes, affect a smaller number of molecular pathways to achieve the classical hallmarks of cancer such as sustained proliferative signaling. Mutation status of several individual pathways was associated with reduced OS in different treatment subgroups. In patients not systemically treated or only treated with endocrine therapy WNT, NOTCH2, p53-independent DNA repair pathway mutation status, and Hedgehog signaling mutation status may identify patients diagnosed as low risk who may benefit from more adjuvant treatment (Fig 6). While these stratification profiles were visible in treatment subgroups, they mostly did not yield significant results in clinical biomarker subgroups (Fig EV3). This may indicate that current risk stratification in histopathological biomarker subgroups is inadequate and should take molecular information into account—something we and others have also shown on the level of gene

expression (Brueffer *et al*, 2018). Identifying the mutation status of pathways and pathway clusters may aid in future clinical trials and treatment, e.g., by aiding selection of treatments that exploit synthetic lethality (Weidle *et al*, 2011).

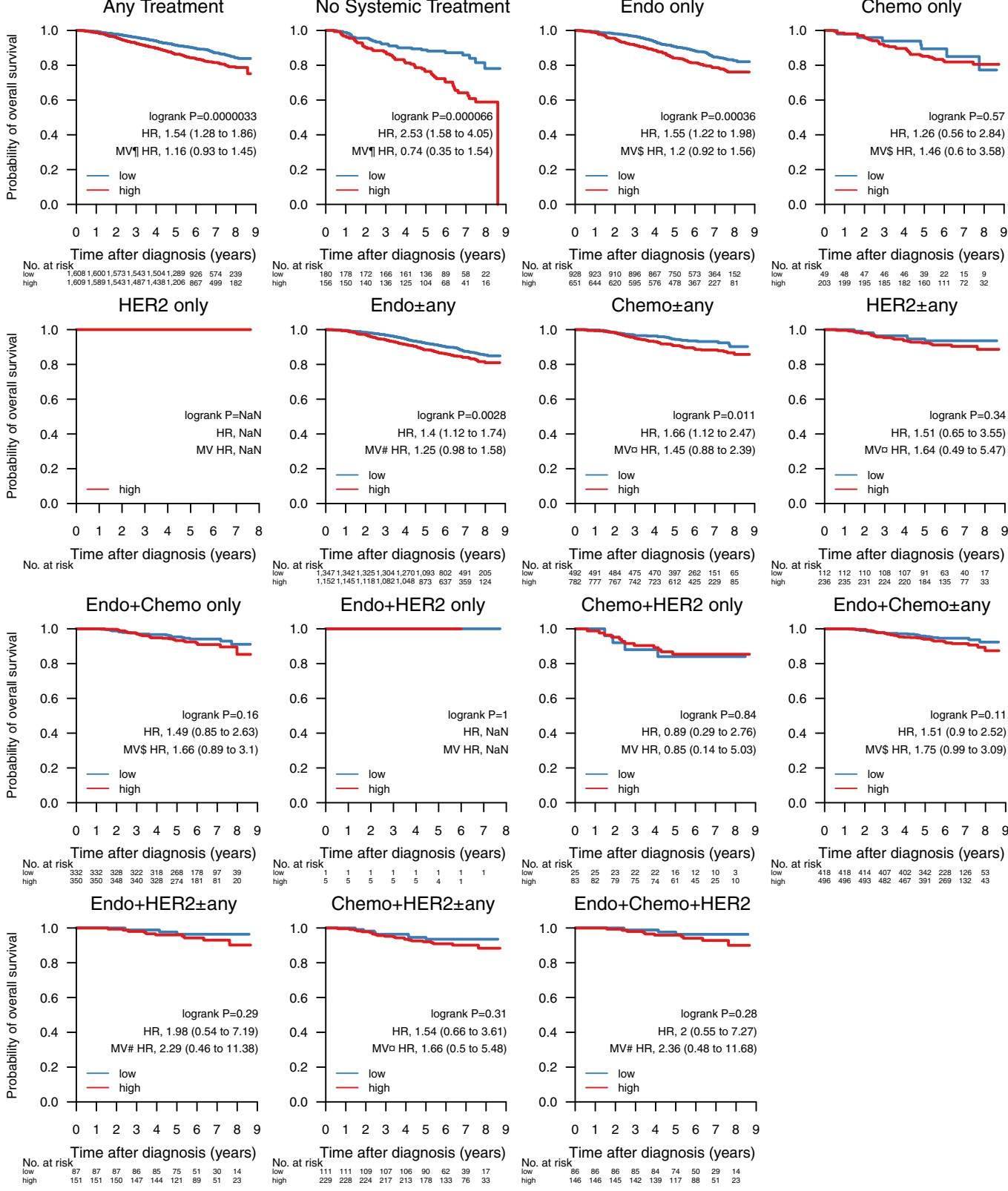

**Figure 7.**

**Figure 7.  Impact of tumor mutational burden on overall survival across treatment groups.**

Overall survival stratified by tumor mutational burden (TMB) across treatment groups in 3,217 patients. Samples were classified as TMB-high if the amount of non-synonymous mutations per expressed MB (rnaMB) was ≥ the median number of non-synonymous mutations per rnaMB across the whole SCAN-B cohort (0.082 mutations per rnaMB) and TMB-low otherwise. In each Kaplan−Meier plot, TMB-low cases are plotted in blue, TMB-high cases are plotted in red, the log-rank *P* value is given, and the hazard ratio (HR) for TMB high is given with a 95% CI and after univariable and multivariable (MV) Cox regression adjustment. Covariables included in the MV analysis were age at diagnosis, lymph node status, tumor size, and the variables denoted by the following symbols: ¶, ER, PgR, HER2, and NHG; ¤, ER, PgR, and NHG; $, HER2 and NHG; #, NHG. ER, estrogen receptor; HER2, human epidermal growth factor receptor 2; HoR, hormone receptor; NHG, Nottingham histological grade; PgR, progesterone receptor; TMB, tumor mutational burden; TNBC, triple-negative breast cancer.

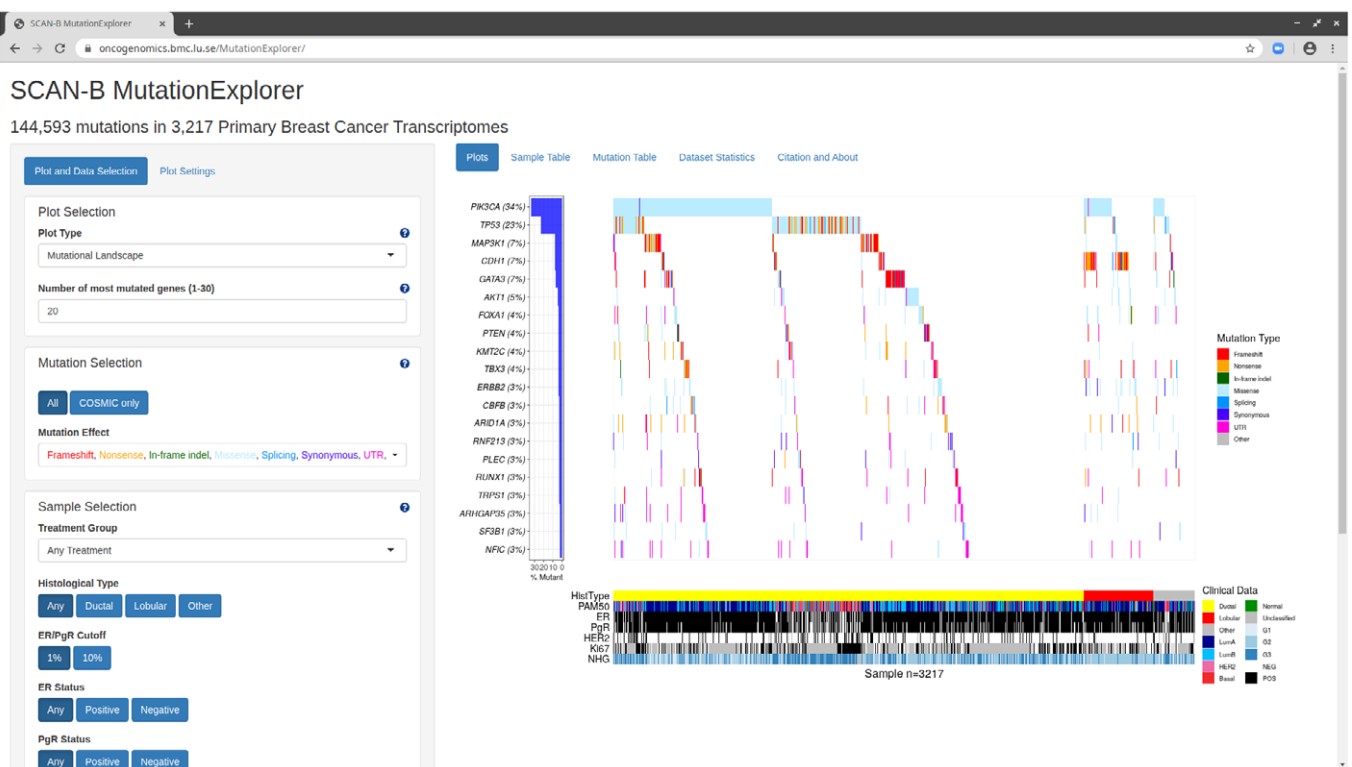

**Figure 8.  The SCAN-B MutationExplorer.**

The SCAN-B MutationExplorer web-based application for interactive exploration of mutations, and their association with clinicopathological subgroups and overall survival. As an example, generation of the image used in Fig 2 is shown.

High TMB has been identified as a predictive biomarker for response to immune checkpoint therapy in diverse solid tumors (Goodman *et al*, 2017; Lauss *et al*, 2017; Hellmann *et al*, 2018; Thomas *et al*, 2018; Zacharakis *et al*, 2018). Using RNA-seq to assess mutational burden may be a useful capability for clinical trials and eventual clinical implementation in BC (Schmid *et al*, 2018). Questions remain however, as TMB is influenced by many biological and technical factors such as ploidy, tumor heterogeneity and clonality (Conroy *et al*, 2019), sample tumor cell content, sequencing depth, and variant filtering. Which cutoff to use for stratifying patients into TMB groups is also still emerging (Panda *et al*, 2017; Schmid *et al*, 2018), and specifically has not been addressed to our knowledge in RNA-seq data. Due to this, and to account for different expression profiles per tumor, we decided to use the median number of non-synonymous mutations per MB of transcriptome across the cohort to stratify patients into TMB-high and TMB-low groups and use it to study OS in different

conventional treatment and biomarker subgroups. In several of these groups, high TMB was significantly associated with worse survival, confirming previous reports (Xu *et al*, 2018), however interestingly not in TNBC. These tumors typically show higher TMB than other clinical BC subtypes, likely because many of them have impaired DNA damage repair mechanisms. Shah and colleagues (Shah *et al*, 2012) showed that only ∼ 36% of mutations in TNBCs are expressed; we speculate that due to this, we may underestimate TMB in several of our TMB-low patients. Additionally, RNA-seq underdetects truncating mutations such as frameshift indels that are a major source of neoantigens. Immune checkpoint therapy is a particularly attractive treatment approach in patients with TNBC and basal-like tumors for which currently no targeted therapy exists. For these patients, determination of TMB using DNA-seq may be a better option than relying on RNA-seq.

Large-scale projects such as TCGA and SCAN-B generate vast amounts of data, but bioinformatics skills are required to make

efficient use of them. Web portals such as cBioPortal (Cerami *et al*, 2012) have emerged to make these huge datasets explorable without specialized skills. In this spirit, we developed the open source web application SCAN-B MutationExplorer to make our mutation dataset easily accessible for other researchers. We hope that SCAN-B MutationExplorer will aid knowledge generation and the development of better BC biomarkers in the future. The open source nature of the portal allows developers to adopt the code for their own purposes, and we welcome contributions of any kind.

### Limitations

The mutation calling we have performed herein tries to achieve sensitive variant calling by using lenient parameters, and heavy filtering of the resulting variants based on stringent quality factors, annotations, and curated databases. This approach has several limitations. While our 275 patient cohort for filter development had matched tumor and normal DNA sequencing data, the SCAN-B cohort only consisted of tumor RNA-seq data. This made accounting for PCR and sequencing artifacts more challenging. Further while many germline events can be filtered by comparing to general databases such as dbSNP, and population-specific ones such as SweGen, these databases are incomplete, and it is thus not possible to remove all germline events this way. As these databases improve, our filters can be upgraded to increase performance. Herein, we also applied filters developed in a matched DNA/RNA set of targeted capture sequencing of 1,697 genes and 1,047 miRNAs (275 sample ABiM cohort) to whole mRNA-seq (3,217 sample SCAN-B cohort). This assumes the transcriptional characteristics of the captured regions are representative for the whole mRNA.

### Conclusion

In summary, we present a tumor-only RNA-seq variant calling strategy and resulting mutation dataset from a large population-based early breast cancer cohort. Although variant calling from RNA-seq data is limited to expressed regions of the genome, mutations in important BC genes such as *PIK3CA*, *TP53*, and *ERBB2*, as well as pathways can be reliably detected, which may be used to inform clinical trials and eventual reporting to the clinic. Mutations in *TP53*, *PIK3CA*, *ERBB2*, and *PTEN* provided prognostic information in several treatment and biomarker patient subgroups, demonstrating the utility of the dataset for research. We make this dataset available for analysis and download via the open source web application SCAN-B MutationExplorer, accessible at http://oncogenomics.bmc.lu.se/MutationExplorer.

## Materials and Methods

### Patients

The study was approved by the Regional Ethics Review Board of Lund at Lund University (diary numbers 2007/155, 2009/658, 2009/659, 2010/383, 2012/58, 2013/459). We analyzed data from two previously described cohorts. For 273 patients, including two

patients with bilateral disease (thus 275 tumors), enrolled in the All Breast Cancer in Malmö (ABiM) study from 2007 to 2009, matched snap-frozen primary breast tumor tissue and blood samples were collected as previously described (Winter *et al*, 2016). A cohort of 3,273 SCAN-B primary breast tumors described previously (Brueffer *et al*, 2018) was reduced to 3,217 samples following additional quality controls. All patients provided informed consent, and the study conforms to the WMA Declaration of Helsinki and the Department of Health and Human Services Belmont Report. Tissue collection, preservation in RNAlater, sequencing, expression estimation, and molecular subtyping using the PAM50 gene list were performed as previously reported (Saal *et al*, 2015; Brueffer *et al*, 2018). Clinical records were retrieved from the Swedish National Cancer Registry (NKBC). Estrogen receptor (ER) and progesterone receptor (PgR) status was categorized using an immunohistochemical staining cutoff of 1%. Patients in the SCAN-B cohort had median 74.5 months follow-up, and patient demographics for both cohorts are detailed in Table 3.

### Library preparation and sequencing

For the 275 sample ABiM cohort, tumor and normal DNA was sequenced using a custom targeted capture panel of 1,697 genes and 1,047 miRNAs as described (Winter *et al*, 2016). For the same tumors, RNA-seq was performed as described (Brueffer *et al*, 2018) (a subset of the 405 sample cohort therein). In short, strand-specific dUTP libraries were prepared and sequenced on an Illumina HiSeq 2000 sequencer to an average of 50 million 101 bp reads per sample (Parkhomchuk *et al*, 2009; Saal *et al*, 2015).

For the 3,217 sample SCAN-B cohort, RNA-seq data were generated as previously described (Brueffer *et al*, 2018). In short, strand-specific dUTP mRNA-seq libraries were prepared (Parkhomchuk *et al*, 2009; Saal *et al*, 2015), and an average 38 million 75 bp reads were sequenced on an Illumina HiSeq 2000 or NextSeq 500 instrument (Table EV1).

### Sequence data processing

For tumor and normal DNA, reads were aligned to the GRCh37 reference genome using Novoalign 2.07.18 (Novocraft Technologies, Malaysia). Using a modified version of the variant workflow of the bcbio-nextgen NGS framework (https://github.com/bcbio/bcbio-nextgen, modified version https://github.com/cbrueffer/bcbio-nextgen/tree/v1.0.2-scanb-calling) utilizing Bioconda for software management (Grüning *et al*, 2018), duplicate reads were marked using biobambam v2.0.62 (Tischler & Leonard, 2014) and variants were called from paired tumor/normal samples using VarDict-Java 1.5.0 (Lai *et al*, 2016) (with default options except -f 0.02 -N ${SAMPLE} -b ${BAM_FILE} -c 1 -S 2 -E 3 -g 4 -Q 10 -r 2 -q 20), which internally performs local realignment around indels. Variant coordinates were converted to the GRCh38 reference genome using CrossMap 2.5 (Zhao *et al*, 2014). Raw RNA-seq reads were trimmed and filtered as described previously (Brueffer *et al*, 2018) and then processed using the modified bcbio-nextgen 1.0.2 variant workflow. Reads were aligned to a version of the GRCh38.p8 reference genome that included alternative sequences and decoys and was patched with dbSNP Build 147 common SNPs, and the GENCODE 25 transcriptome model using HISAT2 2.0.5 (Kim *et al*, 2015) (with default

**Table 3.** Patient demographics and clinicopathological variables in the ABiM and SCAN-B cohorts.

| | ABiM cohort (275 Samples) | | SCAN-B cohort (3,217 Samples) | |
|---|---|---|---|---|
| | Patient count | Percent (%) | Patient count | Percent (%) |
| Age (years) | | | | |
| <50 | 64 | 23.3 | 597 | 18.6 |
| ≥50 | 211 | 76.7 | 2,620 | 81.4 |
| Tumor size (mm) | | | | |
| ≤20 | 145 | 52.7 | 2,080 | 64.7 |
| 21–50 | 120 | 43.6 | 1,018 | 31.6 |
| >50 | 6 | 2.2 | 77 | 2.4 |
| Missing | 3 | 1.1 | 42 | 1.3 |
| Positive lymph nodes (number) | | | | |
| 0 | 151 | 54.9 | 1,974 | 61.4 |
| 1–3 | 60 | 21.8 | 851 | 26.5 |
| ≥4 | 44 | 16.0 | 290 | 9.0 |
| Missing | 20 | 7.3 | 102 | 3.2 |
| Histological type | | | | |
| Ductal | 215 | 78.2 | 2,602 | 80.9 |
| Lobular | 23 | 8.4 | 386 | 12.0 |
| Other | 28 | 10.2 | 229 | 7.1 |
| Missing | 9 | 3.3 | 0 | 0.0 |
| ER status (1% cutoff) | | | | |
| Positive | 223 | 81.1 | 2,786 | 86.6 |
| Negative | 48 | 17.5 | 233 | 7.2 |
| Missing | 4 | 1.5 | 198 | 6.2 |
| PgR status (1% cutoff) | | | | |
| Positive | 204 | 74.2 | 2,509 | 78.0 |
| Negative | 64 | 23.3 | 379 | 11.8 |
| Missing | 7 | 2.5 | 329 | 10.2 |
| HER2 status | | | | |
| Positive | 44 | 16.0 | 414 | 12.9 |
| Negative | 197 | 71.6 | 2,651 | 82.4 |
| Missing | 34 | 12.4 | 152 | 4.7 |
| Nottingham histological grade | | | | |
| Grade 1 | 31 | 11.3 | 483 | 15.0 |
| Grade 2 | 97 | 35.3 | 1,509 | 46.9 |
| Grade 3 | 146 | 53.1 | 1,161 | 36.1 |
| Missing | 1 | 0.4 | 64 | 2.0 |
| Ki67 status | | | | |
| High | 109 | 39.6 | 887 | 27.6 |
| Low | 153 | 55.6 | 627 | 19.5 |
| Missing | 13 | 4.7 | 1,703 | 52.9 |
| Molecular subtype | | | | |
| Luminal A | 109 | 39.6 | 1,545 | 48.0 |
| Luminal B | 83 | 30.2 | 899 | 27.9 |
| HER2-enriched | 30 | 10.9 | 279 | 8.7 |
| Basal-like | 35 | 12.7 | 318 | 9.9 |

**Table 3** (continued)

| | ABiM cohort (275 Samples) | | SCAN-B cohort (3,217 Samples) | |
|---|---|---|---|---|
| | **Patient count** | **Percent (%)** | **Patient count** | **Percent (%)** |
| Normal-like | 13 | 4.7 | 112 | 3.5 |
| Unclassified | 5 | 1.8 | 64 | 2.0 |

options except --rna-strandness RF --rg-id ${ID_NAME} --rg PL:illumina --rg PU:${UNIT} --rg SM:${SAMPLE}). BAM index files were generated using Sambamba 0.6.6 (Faust & Hall, 2014), and duplicate reads were marked using SAMBLASTER 0.1.24 (Tarasov *et al*, 2015). Variants were called using VarDict-Java 1.5.0 with default options except -f 0.02 -N ${SAMPLE} -b ${BAM_FILE} -c 1 -S 2 -E 3 -g 4 -Q 10 -r 2 -q 20 callable_bed, where callable_bed was a sample-specific BED file containing all regions of depth $\geq 4$.

All variants were annotated using a Snakemake (Köster & Rahmann, 2012) workflow around vcfanno 0.3.1 (Pedersen *et al*, 2016) and the data sources dbSNP v151 (Sherry *et al*, 2001), Genome Aggregation Database (gnomAD) (Karczewski *et al*, 2020), Catalogue of Somatic Mutations in Cancer (COSMIC) v87 (Forbes *et al*, 2015; Sondka *et al*, 2018), CIViC (Griffith *et al*, 2017), MyCancerGenome (release March 2016, http://www.mycancergenome.org), SweGen version 20171025 (Ameur *et al*, 2017), the Danish Genome Project population reference (Maretty *et al*, 2017), RNA editing databases (Kiran & Baranov, 2010; Ramaswami & Li, 2014; Sun *et al*, 2016; Picardi *et al*, 2017), UCSC low complexity regions, IntOGen breast cancer driver gene status (Gonzalez-Perez *et al*, 2013) (accessed 2018-08-02), and the drug gene interaction database (DGIdb) v3.0.2 (Cotto *et al*, 2017). We used SnpEff v4.3.1t (with default parameters except hg38 -t -canon) (Cingolani *et al*, 2012b) to predict functional variant impact on canonical transcripts as defined by SnpEff.

To filter out recurrent artifacts introduced during library preparation or sequencing, we constructed a panel of "normal" tissues consisting of all variants enumerated from RNA-seq analysis of adjacent non-tumoral breast tissues sampled from 10 SCAN-B patients.

Gene expression data in fragments per kilobase of transcript per million mapped reads (FPKM) for the ABiM and SCAN-B cohorts were generated as previously reported and is available from the NCBI Gene Expression Omnibus, accession GSE81540 (Brueffer *et al*, 2018).

### Variant filtering

The strategy we applied for developing DNA-seq-informed filters is outlined in Fig 1. Due to the lenient settings used for sensitive initial variant calling, we developed and applied rigid filters to reduce false-positive calls resulting from either sequencing or PCR artifacts, RNA editing, or germline variants. To this end, variants called from 275 matched tumor/normal targeted capture DNA datasets were filtered, among other parameters, for low complexity regions, SNP status (dbSNP "common", SweGen and COSMIC SNPs, high gnomAD allele frequency), allele frequency $\geq 0.05$, depth $\geq 8$, homopolymer environments, and RNA editing sites. Using the resulting DNA variants as reference, we developed filters for the 275 sample RNA-seq variants by permuting values of the sequencing,

variant calling, and annotation variables, and for each permutation calculating the concordance to the DNA mutations. Following these "negative" filters, we applied a range of "positive" filters to rescue filtered variants, e.g., to retain a variant if it is present in the curated MyCancerGenome database of clinically important mutations. Finally, we selected the combination of "negative" and "positive" filter settings with the best balance of sensitivity and specificity. Using SnpSift (Cingolani *et al*, 2012a), we applied the filters to RNA-seq mutation calls from the 3,217 patient cohort. A complete list of final filter variables and values for both the tumor/normal DNA variant calls, as well as the RNA-seq variant calls can be found in Table EV6.

### Data analysis

All analyses were performed using R 3.5.1. Waterfall, heatmap, and lollipop plots were made using the *GenVisR* 1.14.2 (Skidmore *et al*, 2016), *pheatmap* 1.0.12, and *RTrackLayer* 1.42.1 packages. Substitution signatures were analyzed using the *MutationalPatterns* 1.8.0 package (Blokzijl *et al*, 2018). Survival analysis was conducted using OS as endpoint. Overall survival was analyzed using the Kaplan–Meier (KM) method, two-sided log-rank tests, and Cox models, all implemented in the *survival* 2.44-1.1 package. Multivariable Cox models included the variables age at diagnosis, lymph node status, and tumor size as covariables, as well as ER, PgR, HER2, and NHG as relevant. All models were checked for proportional hazards using Grambsch and Therneau's test for non-proportionality and Schoenfeld residuals (Grambsch and Therneau, 1994). Associations were tested using one-tailed and two-tailed Fisher's exact test. *P*-values $< 0.05$ were considered significant. The web application SCAN-B MutationExplorer was written in R using the *Shiny*, *GenVisR*, and *SurvMiner* packages.

## Data availability

The datasets produced and used in this study are available in the following databases:

- Clinical data and mutation calls: http://oncogenomics.bmc.lu.se/MutationExplorer
- Gene expression data: NCBI Gene Expression Omnibus GSE81540 (https://www.ncbi.nlm.nih.gov/geo/query/acc.cgi?acc=GSE81540; Brueffer *et al*, 2018).

Raw patient sequencing data cannot be provided due to Swedish data protection laws.

**Expanded View** for this article is available online.

## The paper explained

### Problem

Breast cancer is a disease of genomic alterations, of which the complete panorama of somatic mutations and how these relate to molecular subtypes, therapy response, and clinical outcomes is incompletely understood. RNA sequencing is a powerful technique for profiling tumor transcriptomes; however, using it for reliable detection of single nucleotide variants and small insertions and deletions is challenging.

### Results

Within the Sweden Cancerome Analysis Network-Breast project (SCAN-B; ClinicalTrials.gov NCT02306096), we developed an optimized bioinformatics pipeline for detection of single nucleotide variants and small insertions and deletions from RNA-seq data. From this, we describe the mutational landscape of 3,217 primary breast cancer transcriptomes and relate it to patient overall survival in a real-world setting (median follow-up 75 months, range 2–105 months). We demonstrate that RNA-seq can be used to call mutations in important breast cancer genes such as *PIK3CA*, *TP53*, *ESR1*, and *ERBB2*, as well as mutation status of key molecular pathways and tumor mutational burden. We identify mutations in one or more potentially druggable genes in 86.8% of cases and reveal significant relationships to patient outcome within specific treatment groups, such as occurrence of mutations inducing resistance to standard of care drugs in untreated patients. To make this rich and growing mutational portraiture of breast cancer available for the wider research community, we developed an open source interactive web application, SCAN-B MutationExplorer, publicly accessible at http://oncogenomics.bmc.lu.se/MutationExplorer.

### Impact

These results add another dimension to the use of RNA-seq as a potential clinical tool, where both gene expression-based signatures and gene mutation-based biomarkers can be interrogated simultaneously and in real-time within 1 week of tumor sampling. Treatment resistance mutations can be detected in early disease and could inform clinical decision-making.

## Acknowledgements

We thank the patients who were part of this study and the SCAN-B study, the employees of the SCAN-B laboratory, the South Sweden Breast Cancer Group, and all SCAN-B collaborators at Hallands Hospital Halmstad, Helsingborg Hospital, Blekinge County Hospital, Central Hospital Kristianstad, Skåne University Hospital Lund/Malmö, Central Hospital Växjö, for inclusion of patients and sampling of tissue for this study. We also thank the Swedish National Breast Cancer Registry and Regional Cancer Center South for clinical data. The SweGen allele frequency data were generated by Science for Life Laboratory. This work was supported by the Mrs. Berta Kamprad Foundation (to ÅB, LHS) and funded in part by the Swedish Research Council (ÅB, LHS), Swedish Cancer Society (ÅB, LHS), Swedish Foundation for Strategic Research (ÅB), Knut and Alice Wallenberg Foundation (ÅB), VINNOVA (ÅB), Governmental Funding of Clinical Research within National Health Service (ALF) (ÅB, LHS), Scientific Committee of Blekinge County Council (AE), Craford Foundation (LHS), Lund University Medical Faculty (LHS), Gunnar Nilsson Cancer Foundation (LHS), Skåne University Hospital Foundation (LHS), BioCARE Research Program (LHS), King Gustav Vth Jubilee Foundation (LHS), and the Krapperup Foundation (LHS).

## Author contributions

CB, CW, and LHS conceived the study. CB, SG, CW, JV-C, JH, AMG, YC, NL, and LHS analyzed data. JV-C, CH, JH, AE, CL, NL, MM, LR, ÅB, and LHS established the SCAN-B initiative. CB, CW, and LHS established the DNA-seq analyses. CB, SG, and LHS established the RNA-seq mutation calling pipeline and filters. JV-C, CH, JH, AE, CL, NL, MM, LR, ÅB, and LHS provided clinical information. LHS supervised the project, and CB and LHS wrote the report with assistance from all authors. All authors discussed, critically revised, and approved the final version of the report for publication.

## Conflict of interest

CB, SG, AMG, YC, and LHS are shareholders and/or employees of SAGA Diagnostics AB. LHS has received honorarium from Novartis and Boehringer-Ingelheim. All remaining authors have declared no conflicts of interest.

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
