## [Review Process File · EMBO Molecular Medicine]

The Mutational Landscape of the SCAN-B Real-World Primary Breast Cancer Transcriptome

Christian Brueffer, Sergii Gladchuk, Christof Winter, Johan Vallon-Christersson, Cecilia Hegardt, Jari Häkkinen, Anthony George, Yilun Chen, Anna Ehinger, Christer Larsson, Niklas Loman, Martin Malmberg, Lisa Rydén, Åke Borg, and Lao Saal

DOI: [10.15252/emmm.202012118](https://doi.org/10.15252/emmm.202012118)

Corresponding authors: Lao Saal (lao.saal@med.lu.se)

Review Timeline:

Submission Date:	2nd Feb 20
Editorial Decision:	27th Feb 20
Revision Received:	9th Jul 20
Editorial Decision:	23rd Jul 20
Revision Received:	8th Aug 20
Accepted:	13th Aug 20

Editor: Lise Roth

Transaction Report:

27th Feb 2020

Dear Dr. Saal,

Thank you for the submission of your manuscript to EMBO Molecular Medicine. We have now received feedback from the two reviewers who agreed to evaluate your manuscript. As you will see from the reports below, the referees acknowledge the interest of the study and are overall supporting publication of your work pending appropriate revisions.

Addressing the reviewers' concerns in full will be necessary for further considering the manuscript in our journal, and acceptance of the manuscript will entail a second round of review. EMBO Molecular Medicine encourages a single round of revision only and therefore, acceptance or rejection of the manuscript will depend on the completeness of your responses included in the next, final version of the manuscript. For this reason, and to save you from any frustrations in the end, I would strongly advise against returning an incomplete revision.

When submitting your revised manuscript, please carefully review the instructions that follow below. Failure to include requested items will delay the evaluation of your revision:

2) Individual production quality figure files as .eps, .tif, .jpg (one file per figure).

3) A .docx formatted letter INCLUDING the reviewers' reports and your detailed point-by-point responses to their comments. As part of the EMBO Press transparent editorial process, the point-by-point response is part of the Review Process File (RPF), which will be published alongside your paper.

4) A complete author checklist, which you can download from our author guidelines (<https://www.embopress.org/page/journal/17574684/authorguide#submissionofrevisions>). Please insert information in the checklist that is also reflected in the manuscript. The completed author checklist will also be part of the RPF.

5) Before submitting your revision, primary datasets produced in this study need to be deposited in an appropriate public database (see <https://www.embopress.org/page/journal/17574684/authorguide#dataavailability>). Please remember to provide a reviewer password if the datasets are not yet public. The accession numbers and database should be listed in a formal "Data Availability" section (placed after Materials & Method). Please note that the Data Availability Section is restricted to new primary data that are part of this study.

6) We would also encourage you to include the source data for figure panels that show essential data. Numerical data should be provided as individual .xls or .csv files (including a tab describing the data). For blots or microscopy, uncropped images should be submitted (using a zip archive if multiple images need to be supplied for one panel). Additional information on source data and instruction on how to label the files are available at

7) Our journal encourages inclusion of *data citations in the reference list* to directly cite datasets that were re-used and obtained from public databases. Data citations in the article text are distinct from normal bibliographical citations and should directly link to the database records from which the data can be accessed. In the main text, data citations are formatted as follows: "Data ref: Smith et al, 2001" or "Data ref: NCBI Sequence Read Archive PRJNA342805, 2017". In the Reference list, data citations must be labeled with "[DATASET]". A data reference must provide the database name, accession number/identifiers and a resolvable link to the landing page from which the data can be accessed at the end of the reference. Further instructions are available at .

8) We replaced Supplementary Information with Expanded View (EV) Figures and Tables that are collapsible/expandable online. A maximum of 5 EV Figures can be typeset. EV Figures should be cited as 'Figure EV1, Figure EV2" etc... in the text and their respective legends should be included in the main text after the legends of regular figures.

- Additional Tables/Datasets should be labeled and referred to as Table EV1, Dataset EV1, etc. Legends have to be provided in a separate tab in case of .xls files. Alternatively, the legend can be supplied as a separate text file (README) and zipped together with the Table/Dataset file. See detailed instructions here:

9) The paper explained: EMBO Molecular Medicine articles are accompanied by a summary of the articles to emphasize the major findings in the paper and their medical implications for the non-specialist reader. Please provide a draft summary of your article highlighting

10) For more information: There is space at the end of each article to list relevant web links for further consultation by our readers. Could you identify some relevant ones and provide such information as well? Some examples are patient associations, relevant databases, OMIM/proteins/genes links, author's websites, etc...

11) Every published paper now includes a 'Synopsis' to further enhance discoverability. Synopses are displayed on the journal webpage and are freely accessible to all readers. They include a short stand first (maximum of 300 characters, including space) as well as 2-5 one-sentences bullet points

that summarizes the paper. Please write the bullet points to summarize the key NEW findings. They should be designed to be complementary to the abstract - i.e. not repeat the same text. We encourage inclusion of key acronyms and quantitative information (maximum of 30 words / bullet point). Please use the passive voice. Please attach these in a separate file or send them by email, we will incorporate them accordingly.

Please also suggest a striking image or visual abstract to illustrate your article. If you do please provide a jpeg file 550 px-wide x 400-px high.

12) As part of the EMBO Publications transparent editorial process initiative (see our Editorial at <http://embomolmed.embopress.org/content/2/9/329>), EMBO Molecular Medicine will publish online a Review Process File (RPF) to accompany accepted manuscripts.

In the event of acceptance, this file will be published in conjunction with your paper and will include the anonymous referee reports, your point-by-point response and all pertinent correspondence relating to the manuscript. Let us know whether you agree with the publication of the RPF and as here, if you want to remove or not any figures from it prior to publication.

I look forward to receiving your revised manuscript.

Yours sincerely,

Lise Roth

Lise Roth, PhD
Editor
EMBO Molecular Medicine

To submit your manuscript, please follow this link:

Link Not Available

***** Reviewer's comments *****

Referee #1 (Comments on Novelty/Model System for Author):

The authors have performed a large scale RNAseq analysis to unravel the mutational landscape of the primary breast cancer transcriptome. The most frequently mutated genes were the known breast cancer drivers PIK3CA, TP53, MAP3K1, CDH1, GATA3 and AKT1. In addition, mutations in relevant molecular pathways were identified including hedgehog signaling, p53-independent DNA repair and hypoxia response pathways. Finally, tumor mutational burden (TMB), a potential biomarker of immunotherapies, was investigated and samples with HER2-enriched and basal-like PAM50 subtypes had the highest TMB values. The mutational landscape was correlated to patient outcome in the total cohort and in known breast cancer subtypes.

Referee #1 (Remarks for Author):

Comments

- (1) This is an excellent work resulting in a very valuable library (SCAN-B MutationExplorer) for the cancer research community. Although the results are not totally novel, the validation of the mutational breast cancer landscape on such a well-defined large patient cohort with a good RNAseq analytical strategy is original and deserves the attention of the research community.
- (2) Although the authors have stratified the groups depending on the type of therapy (Fig. 5 & 6), the patient outcome analyses would profit from multivariate analysis including all significant co-variables to demonstrate which components of the molecular landscape are independent prognosticators and how strong this influence is as compared to the other prognostic factors. When large data sets are analyzed, significant p-values occur even when the effects are minor (e.g., Fig. 5A, Column Endo+Chemo).
- (3) Did any of the patients receive immunotherapy? If yes, the TMB analysis in this cohort would be interesting.
- (4) What kind of endocrine therapy, chemotherapy and anti-HER2 therapies did the patients shown in Fig. 5 & 6 receive?
- (5) Did some patients receive any other targeted therapies (e.g., CDK4/6 inhibitors)?

Referee #2 (Remarks for Author):

The manuscript from Brueffner et al aims to generate mutational landscape from breast cancer transcriptomes. Using matched targeted DNA-seq (tumor+normal) and RNA-seq (tumor only), they implemented a set of filters to perform mutation detection in a large cohort of breast cancer transcriptomes. The results are available on publicly available web portal for exploration.

The manuscript is well written, the aim of the study and the analyses strategy (Section: Variant Filtering, Figure 1) are comprehensible and clearly described.

The manuscript proposes and implements a novel strategy combining a set of rigid filters learnt from DNA-seq and RNA-seq data from matched samples. This is a timely and relevant attempt and worth considering for publication. However, the objective assessment of the results is limited. While the comparison to mutational landscape of TCGA breast cancer data is provided, there is no comparison to alternative strategies currently available (e.g. HaplotypeCaller or Mutect2 combined with filters like dbsnp filter, 1000-genome filter and allele frequency, read depth cut-off, etc.). The

authors should compare their strategy to above mentioned or other available methods and report the results.

The discussion section is disproportionately long and needs to be shortened.

The authors should have their inferred mutation data made available at cbioPortal so that it can be explored in combination with e.g. TCGA and METABRIC datasets. This would further increase the visibility and utility of the dataset.

We thank the Editors and Reviewers for evaluating our manuscript and for the valuable comments and opportunity to submit a revised paper. Below we address the issues raised and incorporate new analyses and revised text which we believe have significantly improved the study. Where possible, we have shortened parts of the paper for succinctness and legibility. We hope the revised paper may be judged as having satisfactorily answering the reviewer comments and may be acceptable for publication in *EMBO Molecular Medicine*.

Referee #1 (Remarks for Author):

Comments

(1) This an excellent work resulting in a very valuable library (SCAN-B MutationExplorer) for the cancer research community. Although the results are not totally novel, the validation of the mutational breast cancer landscape on such a well-defined large patient cohort with a good RNAseq analytical strategy is original and deserves the attention of the research community.

We thank the reviewer for the careful reading of our paper and the positive assessments highlighting the value of our study.

(2) Although the authors have stratified the groups depending on the type of therapy (Fig. 5 & 6), the patient outcome analyses would profit from multivariate analysis including all significant co-variables to demonstrate which components of the molecular landscape are independent prognosticators and how strong this influence is as compared to the other prognostic factors. When large data sets are analyzed, significant p-values occur even when the effects are minor (e.g., Fig. 5A, Column Endo+Chemo).

Thank you for bringing up this point. We agree and have performed the multivariable analyses, and amended the manuscript accordingly.

(3) Did any of the patients receive immunotherapy? If yes, the TMB analysis in this cohort would be interesting.

The patients in this study were diagnosed between 2010-2015 and treated in the clinical routine according to Swedish national guidelines. As such, none of the patients included in this study received immunotherapy. We have added this information to the new Supplementary Table S4 which details the specific therapies per therapy, clinical, and biomarker group.

(4) What kind of endocrine therapy, chemotherapy and anti-HER2 therapies did the patients shown in Fig. 5 & 6 receive?

We thank the reviewer for this question, and since readers may have similar queries about all of the treatment groups, we have now added Supplementary Table S4 which details the specific therapies for all therapy, clinical, and biomarker groups in Figures 5-7 as well as for Figures EV2-5. Note that patients may have received more than one treatment of a certain type, so percentages do not add up to 100%. To directly respond to the reviewer question, the distribution of major therapies for the patients in Figures 5 and 6 is as follows:

Endocrine only

- 55% Tamoxifen
- 48.5% Aromatase-inhibitor
- 0.4% GnRH agonist
- 0.1% other endocrine therapy
- 0.9% unspecified endocrine therapy

Endo + Chemo ± any

- 51.3% Aromatase-inhibitor
- 48.6% Tamoxifen
- 1.9% GnRH agonist
- 0.2% other endocrine therapy
- 0.9% unspecified endocrine therapy
- 88.2% Antracycline
- 18.2% Docetaxel
- 1.3% Paclitaxel
- 1.3% other chemotherapy
- 10.8% unspecified chemotherapy
- 25.3% Trastuzumab
- 0.1% unspecified HER2 therapy

HER2 ± any

- 98.3% Trastuzumab
- 1.7% unspecified anti-HER2 therapy
- 40.2% Aromatase Inhibitors
- 28.4% Tamoxifen
- 1.4% GnRH agonist
- 0.3% unspecified endocrine therapy
- 82.8% Antracycline
- 16.7% Docetaxel
- 1.7% Paclitaxel
- 1.1% other chemotherapy
- 14.1% unspecified chemotherapy

(5) Did some patients receive any other targeted therapies (e.g., CDK4/6 inhibitors)?

The patients in the SCAN-B cohort were enrolled between 2010 and 2015. The first CDK4/6 inhibitors in the enrolling hospitals were given from spring 2017, and only in the metastatic setting. This information is included in the new Supplementary Table S4.

Referee #2 (Remarks for Author):

The manuscript from Brueffner et al aims to generate mutational landscape from breast cancer transcriptomes. Using matched targeted DNA-seq (tumor+normal) and RNA-seq (tumor only), they implemented a set of filters to perform mutation detection in a large cohort of breast cancer transcriptomes. The results are available on publicly available web portal for exploration.

The manuscript is well written, the aim of the study and the analyses strategy (Section:

Variant Filtering, Figure 1) are comprehensible and clearly described.

We thank the reviewer for carefully reviewing our study and for the positive remarks.

The manuscript proposes and implements a novel strategy combining a set of rigid filters learnt from DNA-seq and RNA-seq data from matched samples. This is a timely and relevant attempt and worth considering for publication. However, the objective assessment of the results is limited. While the comparison to mutational landscape of TCGA breast cancer data is provided, there is no comparison to alternative strategies currently available (e.g. HaplotypeCaller or Mutect2 combined with filters like dbsnp filter, 1000-genome filter and allele frequency, read depth cut-off, etc.). The authors should compare their strategy to above mentioned or other available methods and report the results.

We agree that our dataset is well suited for a wide variety of workflow comparisons and benchmarking of different variant callers, parameter combinations, and filtering modalities for RNA-seq variant calling, and that such an endeavor would be extremely valuable. Doing so comprehensively for over 3200 datasets, however, would be a major project in and of itself, both in terms of computational resources, analyses, and time, as demonstrated by similar projects performed on DNA-seq data [1-10]. Moreover, the filter settings that we optimized and applied are rather specific to the output generated by VarDict-Java, and thus having a generalized filtering strategy that would be equally applicable to several mutation callers would require repeating the entire project with this goal in mind.

That being said, to give some indication, we have performed a comparison of the variant callers VarDict-Java (used in our manuscript), Varscan, and Mutect2 with similar configuration settings using twenty samples described in our manuscript. We compared the existing tumor/normal DNA-seq calls for these samples with RNA-seq variant calls when only keeping COSMIC mutations, and when removing dbSNP common mutations (-dbSNP) and rescuing COSMIC mutations (+COSMIC). As detailed in Table 1 below, VarDict-Java and Varscan performed similarly, with VarDict-Java identifying comparable or slightly more variants of all classes than Varscan, whereas Mutect2 generally detected fewer somatic and germline SNVs. Thus, we would argue that our chosen software tool, VarDict-Java, performed adequately for our intended purposes, which is supported by a recent benchmark of software for non-matched variant calling [10].

Table 1 Comparison of RNA-seq variant calls in 20 ABiM samples to matched tumor/normal DNA-based calls using the variant callers VarDict-Java, Varscan, and Mutect2.

Software	Filter	Somatic	Germline	Unknown	Total
Varscan	baseline	141	14109	28000	42250
	+COSMIC	28	9289	11528	20845
	-dbSNP+COSMIC	19	76	683	778
VarDict-Java	baseline	139	15366	29463	44968
	+COSMIC	27	10111	11676	21814
	-dbSNP+COSMIC	19	78	690	787
Mutect2	baseline	104	14235	42779	57118
	+COSMIC	24	8939	10237	19200
	-dbSNP+COSMIC	17	64	1103	1184

Applying the complex filter settings used in the manuscript to calls from either Varscan or Mutect2 is not easily possible, since each caller provides different sequence-based annotations, and the

interpretation of fields such as which filters a variant has to clear to be considered a PASS variant differs. This is another challenge that would be more appropriate to address in a separate comprehensive benchmarking project.

For the reasons outlined above, we respectfully conclude that a systematic comparison to alternative strategies currently available is beyond the scope of the present work. In a future study we intend to more comprehensively compare various software and strategies.

The discussion section is disproportionately long and needs to be shortened.

We agree that the discussion was too long. To address this, we have moved some text blocks to more appropriate places in the manuscript and have merged and removed several paragraphs, as well as overall tightened up the manuscript text.

The authors should have their inferred mutation data made available at cBioPortal so that it can be explored in combination with e.g. TCGA and METABRIC datasets. This would further increase the visibility and utility of the dataset.

We agree that making this dataset easily comparable to other large cohorts such as TCGA via cBioPortal would increase its value, and encouraged by the reviewer's comment, we have initiated the process of submitting the dataset to the cBioPortal team (<https://github.com/cBioPortal/datahub/issues/1101>). However, there are a large number of study cohorts currently awaiting inclusion in cBioPortal, and therefore the availability of our dataset in cBioPortal may take some time and has not been completed in time for the submission of this revision. As soon as this dataset is available in cBioPortal, it will be searchable there, and we will also provide links to this cBioPortal resource from our MutationExplorer site. Furthermore, we shall publicize this availability in our presentations and on social media, as well as (assuming the paper is published) inquire with the editors at EMBO Molecular Medicine whether a link can be provided from the online version of the paper.

References

- [1] Alioto TS, et al. A comprehensive assessment of somatic mutation detection in cancer using whole-genome sequencing. *Nat Commun.* 2015;6
- [2] Xu H, et al. Comparison of somatic mutation calling methods in amplicon and whole exome sequence data. *BMC Genomics.* 2014;15:244.
- [3] Kroigard AB, et al. Evaluation of nine somatic variant callers for detection of somatic mutations in exome and targeted deep sequencing data. *PLoS One.* 2016;11(3):e0151664.
- [4] Wang Q, et al. Detecting somatic point mutations in cancer genome sequencing data: a comparison of mutation callers. *Genome Med.* 2013;5(10):91.
- [5] Roberts ND, et al. A comparative analysis of algorithms for somatic SNV detection in cancer. *Bioinformatics.* 2013;29(18):2223–30.
- [6] Cai L, et al. In-depth comparison of somatic point mutation callers based on different tumor next-generation sequencing depth data. *Sci Rep.* 2016;6:36540.
- [7] Shi et al. Reliability of Whole-Exome Sequencing for Assessing Intratumor Genetic Heterogeneity. *Cell Reports.* 2018;25(6)
- [8] Hofmann et al. Detailed simulation of cancer exome sequencing data reveals differences and common limitations of variant callers. *BMC Bioinformatics.* 2017;18(1)
- [9] Ellrott et al. Scalable Open Science Approach for Mutation Calling of Tumor Exomes Using Multiple Genomic Pipelines. *Cell Systems.* 2018;6(3)
- [10] Sandmann et al. Evaluating Variant Calling Tools for Non-Matched Next-Generation Sequencing Data. *Scientific Reports.* 2017;7:43169

23rd Jul 2020

Dear Dr. Saal,

Thank you for the submission of your revised manuscript to EMBO Molecular Medicine. We have now received the enclosed reports from the two referees who reviewed the new version of your manuscript. As you will see, they are now supportive of publication, and I am thus pleased to inform you that we will be able to accept your manuscript pending the following final editorial amendments:

1) Main manuscript text:

- Please answer/correct the changes suggested by our data editors in the main manuscript file (in track changes mode). This file will be sent to you in the next couple of days. Please use this file for any further modification. Please accept your previous changes, but keep the modifications in response to the data editors in track changes mode.
- Please provide up to 5 keywords.
- Funding: the grant numbers and recipients should be added to the manuscript and in the submission system.
- Material and methods: we note that you regularly refer to previously published methods. Please make sure that sufficient details are provided to ensure reproducibility, and that the referenced literature is open access. Please include a statement confirming that informed consent was obtained from all subjects and that the experiments conformed to the principles set out in the WMA Declaration of Helsinki and the Department of Health and Human Services Belmont Report.
- The Data availability section should only list the accession numbers for the datasets produced in this study, and these accession numbers/links should also be listed in the checklist.
- Several figures indicate logrank $P < 0.01$. When possible, please make sure to indicate the exact p values.

2) Figures:

- Tables 1, 2 and 4 can be directly added to the main manuscript file together with their legend; table 3 should be made an EV Table and stay uploaded as a separate file.
- Fig. 3A should be referenced before Fig. 4 in the manuscript text. References for Fig. 5, Fig. EV2, Fig. EV3 and Fig EV. 4 are missing in the main text.
- All EV table legends should be removed from the manuscript and added directly to the respective files. The EV figure legends should stay in the manuscript, following the main figure legends.
- All figures should fit into an A4 portrait format (i.e. figures 1 and 2), and the text should remain readable.

3) Checklist: The checklist should include the full sentence that informed consent was obtained from all subjects and that the experiments conformed to the principles set out in the WMA Declaration of Helsinki and the Department of Health and Human Services Belmont Report. Accession numbers should be included in section F/18 "Data accessibility".

Regarding the section F/20 on human clinical and genomic datasets to be registered (i.e. dbGAP, EGA), you wrote:

"Gene expression data is available from the NCBI Gene Expression Omnibus (accession Series GSE81540). Mutational and clinical data is available from the SCAN-B Mutation Explorer portal. Raw patient sequencing data cannot be provided due to Swedish data protection laws."

Could you please be more specific? (i.e. what is exactly forbidden by the law? If it is about controlling who can access the data, this could be taken care of by EGA)

4) Thank you for providing a synopsis. I slightly modified your text to fit our style and format, please let me know if you agree with the following:

"A bioinformatics pipeline was developed for detection of single nucleotide variants and small insertions / deletions from RNA sequencing (RNA-seq) data. The mutational landscape of 3,217 primary breast cancer transcriptomes in relation to patient survival was made available through a public web portal.

- An optimized pipeline for detection of single nucleotide variants and short insertions and deletions from RNA-seq data was developed and applied to 3,217 primary breast tumors.
- The mutational portraits identified mutations in clinically important genes, including mutations in one or more potentially druggable genes in 85.3% percent of cases.
- Mutational portraits revealed significant relationships to patient outcome within specific treatment groups, including treatment resistance mutations.
- This rich dataset was made publicly available via our open source web-based application, the SCAN-B MutationExplorer, accessible at <http://oncogenomics.bmc.lu.se/MutationExplorer>."

Please also suggest a striking image or visual abstract to illustrate your article, as a jpeg file 550 px-wide x 400-px high.

5) Thank you for providing "The Paper Explained". Please include it in the main manuscript file, before the references.

6) For more information: There is space at the end of each article to list relevant web links for further consultation by our readers. Some examples are patient associations, relevant databases, OMIM/proteins/genes links, etc...

7) As part of the EMBO Publications transparent editorial process initiative (see our Editorial at <http://embomolmed.embopress.org/content/2/9/329>), EMBO Molecular Medicine will publish online a Review Process File (RPF) to accompany accepted manuscripts.

In the event of acceptance, this file will be published in conjunction with your paper and will include the anonymous referee reports, your point-by-point response and all pertinent correspondence relating to the manuscript. Let us know whether you agree with the publication of the RPF and as here, if you want to remove or not any figures from it prior to publication.

I look forward to receiving your revised manuscript.

Yours sincerely,

Lise Roth

Lise Roth, PhD
Editor
EMBO Molecular Medicine

To submit your manuscript , please follow this link:

Link Not Available

The system will prompt you to fill in your funding and payment information. This will allow Wiley to send you a quote for the article processing charge (APC) in case of acceptance. This quote takes into account any reduction or fee waivers that you may be eligible for. Authors do not need to pay any fees before their manuscript is accepted and transferred to our publisher.

***** Reviewer's comments *****

Referee #1 (Remarks for Author):

The authors have addressed my comments. I have no further remarks.

Referee #2 (Remarks for Author):

The authors have addressed all my comments. The manuscript is suitable for publication at EMM.

The authors performed the requested changes.

The editor accepted the manuscript for publication.

Corresponding Author Name: Lao H Saal, MD PhD

Manuscript Number: EMM-2020-12118